# VFDiff: SE(3)-Equivariant Vector Field Guided Diffusion Model for Target Aware Molecule Generation in 3D

## Abstract

Structure-based drug design (SBDD) is a key challenge in drug discovery that aims to generate small molecules capable of binding tightly to specific protein pockets. However, current diffusion models have focused on the complementarity of ligand molecules and protein pockets in physical space while ignoring the docking energy requirements, resulting in only generating suboptimal docking postures. In this paper, we present VFDiff, a novel SE(3)-equivariant diffusion model for 3D molecular generation, guided by vector fields derived from protein-ligand binding energy. In contrast to current diffusion models, VFDiff incorporates energy-based guidance in both forward and reverse processes to ensure ligand molecules are spatially complementary and energetically matched to their target pockets. Our approach includes three fundamental mechanisms: energy-planning, which adjusts diffusion trajectories based on energy gradients; force-guiding, which refines molecular generation; and position-tuning, which improves sampling accuracy. Extensive experiments on the CrossDocked2020 dataset demonstrate that VFDiff outperforms state-of-the-art methods, achieving superior binding binding affinity with an impressive Avg. Vina Score of up to **-7.37**, while maintaining competitive molecular properties, and diversity. This work introduces a new framework for generating target-specific molecules with improved structural and functional fidelity, offering a significant advancement in SBDD.

## 1 Introduction

Structure-based drug design (SBDD) technology is one of the economical and efficient means to address the development of lead compounds against specific protein pocket targets (Anderson (2003); Batool et al. (2019)). The goal of SBDD is to generate small molecules that are structurally complementary to the pocket space and matched in energy (Kalyaanamoorthy & Chen (2011)), thus enabling tighter binding of the molecule to the pocket and exerting the drug function of the molecule.

Recently, with rapid progress in structural biology and the increasing scale of structural data (Francoeur et al. (2020)), several target-aware generative methods have achieved remarkable results in this field (Luo et al. (2021); Liu et al. (2022); Peng et al. (2022); Huang et al. (2024b); Lin et al. (2022)). In particular, equivariant diffusion models (EDMs) have emerged as the current SOTA model in molecular generation, excelling in preserving geometric symmetry in the marginal distribution of generated molecules (Hoogeboom et al. (2022); Xu et al. (2022); Guan et al. (2023; 2024); Bao et al. (2022)). Nevertheless, we found that these models were far from our desired goal, *i.e.*, generating ligand molecules that are complementary to the protein pocket shape and energetically matched. Specifically, what these EDMs do is actually to reconstruct the trajectory of a predefined stochastic process. For different ligands in the complexes, the same noise-added disturbing strategy actually just allowed the model generate ligand molecules spatially compatible with the protein pockets but failed to show control on the energy matching. Therefore, we believe that the current EDM will limit the molecules' properties in this regard.

To bridge the gap between existing SBDD models and the necessity for designing ideal ligands, we propose the **V**ector **F**ield Guided **Diff**usion (**VFDiff**), a shifted-diffusion model with guidance

of binding energy. Specifically, we first train an SE(3)-equivariant neural network called **VFNet** on PDBBind2016 (Wang et al. (2004)) by supervised learning to learn the energy function for improving the controllability of generation. In physics, the derivation of the energy score to atomic coordinates will yield a force field (instead of vector field in the later text), which naturally exploits the geometric symmetry in 3D molecular conformation, as long as the energy function is invariant to orthogonal transformations. This vector field reflects the force on the ligand molecule in the protein pocket and also indicates the direction of the energy decrease (Pissurlenkar et al. (2009)). We introduce this prior information into our diffusion model with an elaborated schedule such that the endpoint of the modified trajectory remains practically unchanged compared to the final distribution of a standard diffusion process. In the forward process, the shifted-prior predicted by VFNet would cause the molecules to diffuse in the direction of decreasing energy (**energy-planning**) and in the reverse process guide the molecules toward increasing energy (**force-guiding**). In addition another vector field predicted by VFNet will fine-tune the initial position predicted in the inverse process to obtain more accurate predictions of atom coordinates (**position-tuning**). To demonstrate the efficacy of our VFDiff, we conduct extensive experiments on the CrossDocked2020 (Francoeur et al. (2020)) dataset. Empirical results show that our VFDiff can generate ligands that not only bind tightly to target pockets but also maintain proper molecular properties, outperforming existing diffusion-based molecular models (The code of VFDiff will be publicly accessible upon acceptance).

We highlight our contributions as follows:

• We propose a novel 3D geometric diffusion model (VFDiff) for SBDD, which facilitates ligand generation by controlling changes in pocket-ligand binding affinity during the forward and reverse processes. Additionally, we ensure the equivariance of coordinate transformations throughout these processes.

• We design position-tuning to enhance sampling accuracy in the reverse process by fine-tuning the coordinates of the predicted $\hat{X}_0$, helping the model establish correct molecular conformations early in the generation process.

• VFDiff achieves state of the art (SOTA) performance on CrossDocked2020 benchmark, and it can generate the molecules with **-7.37** Avg. Vina Score while maintaining proper molecular properties.

## 2 RELATED WORK

**Structure-based Drug Design** Anderson (2003) focuses on generating ligand molecules that bind to a specific target protein. Skalic et al. (2019); Xu et al. (2021) proposed generating ligand molecules in SMILES format conditioned on protein contexts. Tan et al. (2023) developed a flow model capable of generating validated molecular drugs as 2D graphs based on the sequence embeddings of specific targets. Ragoza et al. (2022) voxelized molecules into atomic density grids and applied VAE to generate 3D ligand molecules at receptor binding sites. Luo et al. (2021); Liu et al. (2022); Peng et al. (2022) introduced a method for generating atoms and bonds in 3D Euclidean space in an autoregressive manner. Zhang et al. (2023) proposed generating 3D ligand molecules fragment by fragment. Recently, diffusion models have been successfully applied to SBDD, showing promising results, which will be discussed in more detail in the following section. Our work aims to enhance 3D molecular diffusion models for SBDD.

**Diffusion Models for SBDD** Guan et al. (2023); Lin et al. (2022); Schneuing et al. (2022) utilize diffusion models to first generate atom types and positions, followed by defining the bonds in a post-processing step. They propose using SE(3)-equivariant neural networks to denoise ligand molecules within the protein-ligand complex, where the protein pocket remains fixed. Schneuing et al. (2022) additionally attempts to generate compounds (i.e., the protein-ligand complex) by inpainting conditioned on the binding site of the pocket. Guan et al. (2024) further incorporates decomposed priors into diffusion models for SBDD, inspired by traditional drug discovery methods, and achieves high average binding affinity. Binding affinity, an essential metric in drug discovery, evaluates whether drugs selectively and specifically bind to their target proteins. While DecompDiff Guan et al. (2024) has delivered promising results, it depends heavily on external computational tools, such as AlphaSpace2 Rooklin et al. (2015), to extract subpockets and generate pocket priors when creating ligand molecules for new pockets. IPDiff sought to integrate protein-ligand interactions into diffusion model generation, but it failed to incorporate explicit modeling of these interactions in 3D coor-

dinates for direct and accurate guidance, and the lack of geometric symmetry guided information limited the model's performance. Different from all the above, our molecular diffusion model for the first time considers pockets, ligand molecules, and their SE(3)-equivariant energy information in both forward and reverse processes. Moreover, our VFDiff has no dependency on any external tools and outperforms all other methods in binding affinity, bond distance distribution, and in diversity.

## 3 PRELIMINARY

In this section we will define the SBDD task from the perspective of generative modeling, a task where the goal is that given protein pocket information we expect to generate high-affinity conformations of that pocket-bound small molecule by modeling. The target (protein pocket) and ligand molecule can be represented as $\mathcal{P} = \{(\mathbf{x}_i^{\mathcal{P}}, \mathbf{v}_i^{\mathcal{P}})\}_{i=1}^{N_{\mathcal{P}}}$ and $\mathcal{M} = \{(\mathbf{x}_i^{\mathcal{M}}, \mathbf{v}_i^{\mathcal{M}})\}_{i=1}^{N_{\mathcal{M}}}$, respectively. Here, $N_{\mathcal{P}}$ (resp. $N_{\mathcal{M}}$) denotes the number of atoms in the protein pocket $\mathcal{P}$ (resp. the ligand molecule $\mathcal{M}$). $\mathbf{x}_i^{\mathcal{P}} \in \mathbb{R}^3$ represents the 3D coordinates of the protein atom, and $\mathbf{v}_i^{\mathcal{P}} \in \mathbb{R}^{N_f}$ represents protein atom features such as element types and amino acid types, with $N_f$ representing the number of such features. In this work, we aim to optimize binding affinity, by generating prospective ligand molecules $\mathcal{M}$, for a given protein $\mathcal{P}$. Here, $\mathbf{x}_i^{\mathcal{M}} \in \mathbb{R}^3$ and $\mathbf{v}_i^{\mathcal{M}} \in \mathbb{R}^K$ represent the atom coordinates and atom types of a ligand molecule, respectively. The variable $K$ indicates the number of features used to characterize these atom types. The variable $N_{\mathcal{M}}$ signifies the number of atoms in the ligand molecule, which can be sampled during inference utilizing either an empirical distribution or predicted through a neural network, and the chemical bonds are generated by the post-processing programs. We define $\beta_t$ $(t = 1, ..., T)$ as fixed variance schedules, with $\alpha_t = 1 - \beta_t, \bar{\alpha}_t = \prod_{s=1}^{t} \alpha_s, \bar{\beta}_t = 1 - \bar{\alpha}_t$.

To simplify, we use matrix representation to denote the ligand molecule as $\mathbf{M} = [\mathbf{X}^{\mathcal{M}}, \mathbf{V}^{\mathcal{M}}]$, where $\mathbf{X}^{\mathcal{M}} \in \mathbb{R}^{3 \times N_{\mathcal{M}}}$ and $\mathbf{V}^{\mathcal{M}} \in \mathbb{R}^{N_{\mathcal{M}} \times K}$. Similarly, we represent the protein is denoted as $\mathbf{P} = [\mathbf{X}^{\mathcal{P}}, \mathbf{V}^{\mathcal{P}}]$, where $\mathbf{X}^{\mathcal{P}} \in \mathbb{R}^{3 \times N_{\mathcal{P}}}$ and $\mathbf{V}^{\mathcal{P}} \in \mathbb{R}^{N_{\mathcal{P}} \times N_f}$.

## 4 METHOD

As described in the previous sections, we attempted to introduce information about the binding energy of protein-ligand complexes into the forward and backward processes of the diffusion model to advance molecule generation. Therefore, we propose VFDiff, a novel diffusion model based on vector field guidance. We first design a prior network VFNet to capture the interactions between pockets and ligands from the perspective of both 3D geometric information and chemical properties, and pretrain it with binding affinity signals and noised 3D coordinates. Then, we take the pretrained VFNet as a provider of binding affinity information to facilitate the binding-aware ligand diffusion process. To fully utilize protein-molecule binding information in both forward and reverse processes of our diffusion framework, we introduce three mechanisms: energy-planning, force-guiding, and position-tuning (detailed in Section 4.2).

### 4.1 LEARNING PROTEIN-LIGAND BINDING INFORMATION PRIOR WITH VFNET

VFNet consists of $L$ layers SE(3)-equivariant neural networks. These SE(3)-equivariant neural networks are applied on the complex graph $\mathcal{G}^{\mathcal{C}} = \mathcal{G}^{\mathcal{M}} \cup \mathcal{G}^{\mathcal{P}}$, which is constructed by $\mathcal{G}^{\mathcal{M}}$ and $\mathcal{G}^{\mathcal{P}}$, to model the forces acting on the ligand molecule within the protein pocket. For a given complex graph $\mathcal{G}^{\mathcal{C}}$, the $l$-th SE(3)-equivariant layer update the atom embeddings h and coordinates x as follow:

$$\mathbf{h}_i^{l+1} = \mathbf{h}_i^l + \sum_{j \in \mathcal{N}_C(i), i \neq j} f_h(d_{ij}^l, \mathbf{h}_i^l, \mathbf{h}_j^l, \mathbf{e}_{ij})$$

$$\mathbf{x}_i^{l+1} = \mathbf{x}_i^l + \sum_{j \in \mathcal{N}_C(i), i \neq j} (\mathbf{x}_i^l - \mathbf{x}_j^l) f_x(d_{ij}^l, \mathbf{h}_i^{l+1}, \mathbf{h}_j^{l+1}, \mathbf{e}_{ij}) \cdot \mathbb{1}_{\text{mol}} \tag{1}$$

where $d_{ij} = \| \mathbf{x}_i - \mathbf{x}_j \|$ is the euclidean distance between atoms $i$ and $j$. The variable $\mathbf{e}_{ij}$ is an additional feature indicating the connection is between protein atoms, ligand atoms or protein atom

Figure 1: The overall schematic diagram of VFDiff. The pretrained VFNet are frozen during both training and sampling process for providing interaction priors. The molecule $\mathbf{M}_0$ and $\hat{\mathbf{M}}_{0|t}$ are utilized for extracting energy information in forward and reverse process, respectively. The $\hat{\mathbf{M}}_{0|t}$ is the estimated molecule at the time step $t$ of the sampling process due to the inaccessibility of $\mathbf{M}_0$, $\mathbf{F}$ denotes the derived vector field of molecules.

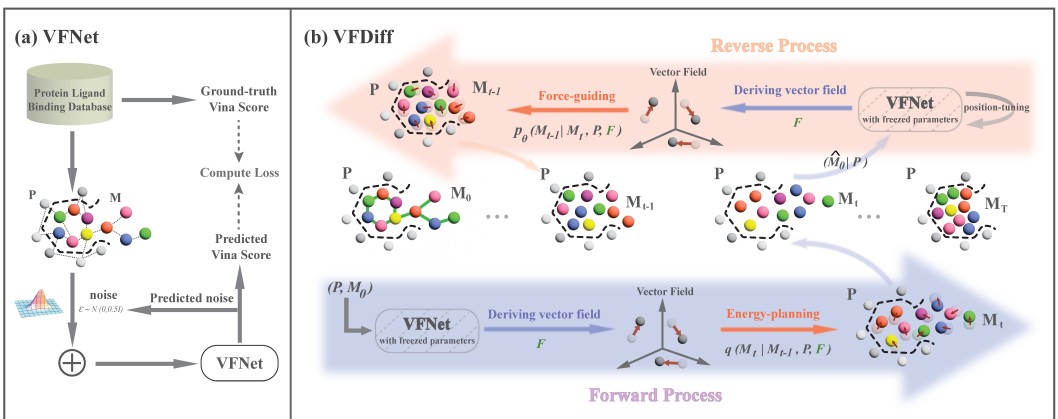

and ligand atom. $\mathcal{N}_C(i)$ stands for the set of neighbors of atom $i$ on $\mathcal{G}^{\mathcal{C}}$, and $\mathbb{1}_{\mathrm{mol}}$ is the ligand molecule mask since we do not want to update protein atom coordinates.

After $L$ layers of neural networks, we obtained the final coordinates $\mathbf{X}^L$, which would be used to model *Geometry-to-Energy mapping*. At the beginning of the generation phase, the predicted molecular conformation may be unstable, leading to inaccuracies in the vector field inferred by VFNet from the binding energy. Therefore, VFNet needs to be adapted to this unstable structure, so we adjusted the binding energy score of the complex with added fine noise and the loss functions are shown below (more analysis about adjusted score can be found in Appendix D.2):

$$\hat{y} = \sum_{i \in N_{\mathcal{M}}} \left\| \mathbf{x}_i^L - \mathbf{x}_i^0 \right\|_2^2, \quad \mathbf{x}_i^0 = \mathbf{x}_i^{\mathcal{M}} + \eta_1 \cdot \epsilon,$$

$$\mathcal{L} = \mathbb{E}_{\mathbf{X} \sim \mathcal{D}} \left[ \mathbb{E}_{\epsilon \sim \mathcal{N}(0,1), \eta_1 \sim \mathcal{U}(0,0.5)} \left[ \| \frac{y}{1+\eta_1} - \hat{y} \|_2^2 \right] \right], \tag{2}$$

where dataset $\mathcal{D}$ contains protein-ligand pairs' information, $\mathbf{X} = \mathbf{M} \cup \mathbf{P}$, $\epsilon$ is a noise disturbance, and $\eta_1$ is a scaling coefficient, which would be set to $0$ during inferencing. It is evident that the predicted energy score $\hat{y}$ is an invariant scalar with respect to the aforementioned derivation. By differentiating it with respect to $\mathbf{X}^{\mathcal{M}}$, we obtain the direction in which the binding energy increases, known as the force field $\mathbf{F}$. This force field can be formulated as follows:

$$\mathbf{F} = -\nabla_{\mathbf{X}^{\mathcal{M}}} \hat{y} \tag{3}$$

The force field $\mathbf{F}$, obtained by differentiating the scalar $\hat{y}$ with respect to the ligand atomic coordinates $\mathbf{X}^{\mathcal{M}}$, is SE(3)-equivariant if $\hat{y}$ remains SE(3)-invariant under a orthogonal transformation $f(\cdot)$ with respect to $\mathbf{X}^{\mathcal{M}}$. Additionally, VFNet can also be utilized as a coordinate tuner to correct the predicted atomic coordinates $\hat{\mathbf{x}}_{i,0}^{\mathcal{M}}$ at $t = 0$ by VFDiff during the initial stage of the reverse generation process. Experimental results demonstrate that this correction plays a significant role in fitting the prior distribution of atomic bond lengths and controlling molecular properties.

To help the diffusion model obtain accurate molecular conformations in the early stages of sampling, we trained a new VFNet from scratch on slightly noised data without changing the model architecture, enabling it to acquire position-tuning capabilities. Similar to training a score-based model (SBM), we expect the model to output the initial atomic direction (like score in SBM). The training objective for position-tuning can be expressed by the following equation:

$$\mathcal{L} = \mathbb{E}_{\mathbf{X}\sim\mathcal{D}}\left[\mathbb{E}_{\eta_2\sim\mathcal{U}(0,0.5),\mathbf{E}\sim\mathcal{N}(\mathbf{0},\mathbf{I})}\left[\text{Sim}\big(f_\psi(\mathbf{X}'_{\mathcal{M}},\mathbf{X}^{\mathcal{P}},\mathbf{V}^{\mathcal{M}},\mathbf{V}^{\mathcal{P}})-\mathbf{X}'_{\mathcal{M}},\mathbf{E}\big)\right]\right] \quad (4)$$

where $\mathbf{X} = \mathbf{X}^{\mathcal{M}}\cup\mathbf{X}^{\mathcal{P}}$, $\mathbf{X}'_{\mathcal{M}} = \mathbf{X}^{\mathcal{M}}+\eta_2\cdot\mathbf{E}$, $\mathbf{E}$ has the same shape as $\mathbf{X}^{\mathcal{M}} \in \mathbb{R}^{3\times N_{\mathcal{M}}}$ and is a single sample from a standard multivariate normal distribution, $\eta_2$ is a scaling coefficient sampled from a uniform distribution, and $f_\psi(\cdot)$ is the output of the last layer of molecular coordinates $\mathbf{x}^l$ in VFNet. $\text{Sim}(\cdot,\cdot)$ denotes the cosine similarity. We anticipate that VFNet will learn the true distribution of atomic coordinates in the perturbed data so that the $\hat{\mathbf{X}}_0^{\mathcal{M}}$ samples early in the generation process will fall where the data density is higher and will enhance the accuracy of the molecular conformation. Next, we will describe how to utilize the prior network VFNet to facilitate the 3D molecular diffusion generation with vector field in both the forward and reverse processes.

## 4.2 VECTOR FIELD GUIDED 3D MOLECULAR DIFFUSION MODEL

### 4.2.1 SE(3)-EQUIVARIANT ENERGY-PLANNING

We proposed energy-planning makes the trajectory of the ligand molecule distribution towards in the direction of the fastest binding energy decrease based on VFNet predictions. Unlike conventional diffusion models where the forward process simply tries to break the original distribution to obtain a normal distribution, our forward process introduces a learned SE(3)-equivariant bias term to make the trajectory physically meaningful. Through a carefully designed schedule, we add different strengths of bias to the samples at different moments in time:

$$\mathbf{S}_t^{\mathcal{M}} = \eta_3 \cdot k_t \cdot \mathbf{F}_t, \tag{5}$$

Next we describe how to compute $\mathbf{F}_t$.

$$q(\tilde{\mathbf{M}}_t|\mathbf{M}_0,\mathbf{P}) = \prod_{i=1}^{N_M}\mathcal{N}(\tilde{\mathbf{x}}_{t,i}^{\mathcal{M}};\sqrt{\bar{\alpha}_t}\mathbf{x}_0+\sqrt{\alpha_t},(1-\bar{\alpha}_t)I)\cdot\mathcal{C}(\mathbf{v}_{0,i}^{\mathcal{M}}|\sqrt{\bar{\alpha}_t}\mathbf{v}_0+(1-\bar{\alpha}_t)/K), \quad (6)$$

where equation 6 is a standard diffusion process and $\tilde{\mathbf{X}}_t^{\mathcal{M}}$ is a sample from this process. $\mathbf{F}_t$ is vector field derived from $\tilde{\mathbf{X}}_t^{\mathcal{M}}$ by a VFNet according to equation 2 and 3, $k_t$ is shifted-mode coefficient and $\eta_3$ is shifted-scale coefficient. Following Huang et al. (2024c), we set $k_t$ to $\sqrt{\bar{\alpha}_t}\cdot(1-\sqrt{\bar{\alpha}_t})$, since we do not wish to change the final distributions of the diffusion process but rather the trajectory of this process. $\eta_3$ is introduced to control the shift scales of diffusion trajectories, and we set $\eta_3 = 10$ in our experiments.

More specifically, at each step of the forward process of VFDiff, we move the molecular conformation one step in the direction of decreasing affinity and then scale it so that the trajectory of the overall distribution change is biased in the direction of the fastest decreasing affinity (the differences from the standard molecular diffusion in Equation 6 are highlighted in purple):

$$q(\mathbf{M}_t|\mathbf{M}_0,\mathbf{P},\mathbf{F}_t) = \prod_{i=1}^{N_M}\mathcal{N}(\mathbf{x}_{t,i}^{\mathcal{M}};\sqrt{\bar{\alpha}_t}\mathbf{x}_0+\sqrt{\alpha_t}\mathbf{s}_{t,i}^{\mathcal{M}},(1-\bar{\alpha}_t)I)\cdot\mathcal{C}(\mathbf{v}_{0,i}^{\mathcal{M}}|\sqrt{\bar{\alpha}_t}\mathbf{v}_0+(1-\bar{\alpha}_t)/K), \quad (7)$$

$$q(\mathbf{M}_t|\mathbf{M}_{t-1},\mathbf{P},\mathbf{F}_t,\mathbf{F}_{t-1}) = \prod_{i=1}^{N_M}\mathcal{N}(\mathbf{x}_{t,i}^{\mathcal{M}};\sqrt{\alpha_t}(\mathbf{x}_{t-1}+\mathbf{s}_{t,i}^{\mathcal{M}}-\sqrt{\alpha_{t-1}}\mathbf{s}_{t-1,i}^{\mathcal{M}}),(1-\alpha_t)I)\cdot$$
$$\mathcal{C}(\mathbf{v}_{0,i}^{\mathcal{M}}|\sqrt{\alpha_t}\mathbf{v}_0+(1-\alpha_t)/K), \tag{8}$$

where $\mathcal{N}$ and $\mathcal{C}$ stand for the Gaussian and categorical distribution, respectively. And $\mathbf{s}_{t,i}^{\mathcal{M}}$ denotes the vector on the $i$-th row of $\mathbf{S}_t^{\mathcal{M}}$. We shall prove these two definitions are consistent in Appendix C.1. It is worth noting that the shifted bias $\mathbf{s}_t$ is SE(3)-equvariant, which allows the important geometric symmetry that satisfied for constructing the probability density distribution of the molecule to be hold (the proof can be found in the appendix C.4):

$$q(\mathbf{X}_t^{\mathcal{M}}|\mathbf{X}_0^{\mathcal{M}},\mathbf{P},\mathbf{F}_t) = q(\mathbf{R}\mathbf{X}_t^{\mathcal{M}}|\mathbf{R}\mathbf{X}_0^{\mathcal{M}},\mathbf{R}\mathbf{P},\mathbf{F}_t'), \tag{9}$$

where $\mathbf{R} \in \mathbb{R}^{3\times3}$ denotes a rotation matrix, $\mathbf{X}^{\mathcal{M}} \in \mathbb{R}^{3\times N_{\mathcal{M}}}$ denotes molecule coordinate and $\mathbf{F}_t'$ is the vector field predicted by VFNet after rotating $\tilde{\mathbf{X}}_t^{\mathcal{M}}$ and $\mathbf{P}$. We do not introduce any other hard-to-solve bias, so equation (7) and (8) can be easily derived by VFNet.

### 4.2.2 FORCE-GUIDING AND POSITION-TUNING

In the previous subsection, we described how to introduce vector fields into a forward procedure. In this subsection we describe how force-guiding and position-tuning guide the molecule generation process during sampling. Using Bayes' formula, the posterior distribution of shifted-diffusion at different moments can be derived as follows:

$$
q(\mathbf{M}_{t-1}|\mathbf{M}_t,\mathbf{M}_0,\mathbf{P},\mathbf{F}_t,\mathbf{F}_{t-1}) = \prod_{i=1}^{N_{\mathcal{M}}} \mathcal{N}\left(\mathbf{x}_{t-1,i}^{\mathcal{M}}; \widetilde{\boldsymbol{\mu}}(\mathbf{x}_{t,i}^{\mathcal{M}},\mathbf{x}_{0,i}^{\mathcal{M}},\mathbf{f}_{t,i}^{\mathcal{M}},\mathbf{f}_{t-1,i}^{\mathcal{M}}), \widetilde{\beta}_t I\right) \cdot
$$
$$
\mathcal{C}\left(\mathbf{v}_{t-1,i}^{\mathcal{M}}|\mathbf{c}(\mathbf{v}_{t,i}^{\mathcal{M}},\mathbf{v}_{0|t,i}^{\mathcal{M}})\right), \tag{10}
$$

where $\widetilde{\boldsymbol{\mu}}(\mathbf{x}_{t,i}^{\mathcal{M}},\mathbf{x}_{0,i}^{\mathcal{M}},\mathbf{f}_{t,i}^{\mathcal{M}},\mathbf{f}_{t-1,i}^{\mathcal{M}}) = \frac{\sqrt{\alpha_t}(1-\bar{\alpha}_{t-1})}{1-\bar{\alpha}_t}\mathbf{x}_t + \frac{\sqrt{\bar{\alpha}_{t-1}}(1-\alpha_t)}{1-\bar{\alpha}_t}\mathbf{x}_0 + \frac{\alpha_t(\bar{\alpha}_{t-1}-1)}{1-\bar{\alpha}_t}\mathbf{s}_t + \sqrt{\alpha_{t-1}}\mathbf{s}_{t-1}$, $\widetilde{\beta}_t = \frac{1-\bar{\alpha}_{t-1}}{1-\bar{\alpha}_t}\beta_t$, $\mathbf{c}(\mathbf{v}_{t,i}^{\mathcal{M}},\hat{\mathbf{v}}_{0|t,i}^{\mathcal{M}}) = \frac{\mathbf{c}^*}{\sum_{k=i}^{K}c_k^*}[\alpha\mathbf{v}_{t,i} + (1-\alpha_t)/K] \odot [\bar{\alpha}_{t-1}\mathbf{v}_{0,i} + (1-\bar{\alpha}_{t-1})/K]$.

In the real sampling process, the ground-truth Molecule $\mathbf{X}_0$ and $\mathbf{V}_0$ is inaccessible at time step $t$, we replace it with the $\hat{\mathbf{M}}_{0|t} = [\hat{\mathbf{X}}_{0|t}, \hat{\mathbf{V}}_{0|t}]$ predicted by a model at time step $t$. After that, we sample $\tilde{\mathbf{X}}_t^{\mathcal{M}}$ based on $\hat{\mathbf{M}}_{0|t}$ with equation 6 and feed it into the VFNet network to obtain the vector field $\mathbf{F}_t$ with equation (1,2,3). Then we can compute the predicted shifted-bias $\hat{\mathbf{s}}_t$ at time step $t$ by using equation (5). Therefore, the reverse transition kernel can be approximated with predicted atom types $\hat{\mathbf{v}}_{0|t,i}$, atom positions $\hat{\mathbf{x}}_{0|t,i}$, and vector field $\hat{\mathbf{f}}_{t,i}^{\mathcal{M}}$ as follows:

$$
p_\theta(\mathbf{M}_{t-1}|\mathbf{M}_t,\mathbf{P},\mathbf{F}_t,\mathbf{F}_{t-1}) = \prod_{i=1}^{N_{\mathcal{M}}} \mathcal{N}\left(\mathbf{x}_{t-1,i}^{\mathcal{M}}; \widetilde{\boldsymbol{\mu}}(\mathbf{x}_{t,i}^{\mathcal{M}},\hat{\mathbf{x}}_{0|t,i}^{\mathcal{M}},\hat{\mathbf{f}}_{t,i}^{\mathcal{M}},\hat{\mathbf{f}}_{t-1,i}^{\mathcal{M}}), \widetilde{\beta}_t I\right) \cdot
$$
$$
\mathcal{C}\left(\mathbf{v}_{t-1,i}^{\mathcal{M}}|\mathbf{c}(\mathbf{v}_{t,i}^{\mathcal{M}},\hat{\mathbf{v}}_{0|t,i}^{\mathcal{M}})\right), \tag{11}
$$

where $\widetilde{\boldsymbol{\mu}}(\mathbf{x}_{t,i}^{\mathcal{M}},\mathbf{x}_{0,i}^{\mathcal{M}},\hat{\mathbf{f}}_{t,i}^{\mathcal{M}},\hat{\mathbf{f}}_{t-1,i}^{\mathcal{M}}) = \frac{\sqrt{\alpha_t}(1-\bar{\alpha}_{t-1})}{1-\bar{\alpha}_t}\mathbf{x}_{t,i} + \frac{\sqrt{\bar{\alpha}_{t-1}}(1-\alpha_t)}{1-\bar{\alpha}_t}\hat{\mathbf{x}}_{0|t,i} + \frac{\alpha_t(\bar{\alpha}_{t-1}-1)}{1-\bar{\alpha}_t}\hat{\mathbf{s}}_{t,i} + \sqrt{\alpha_{t-1}}\hat{\mathbf{s}}_{t-1,i}$. In this way, our VFDiff can align molecular diffusion process with molecular sampling process regarding the information utilization of target protein, and guiding the diffusion trajectories according to the energy landscape.

As previously expressed, the affinity gradient of ligand is calculated based on the sampling $\tilde{\mathbf{X}}_t^{\mathcal{M}}$, which is derived from $\hat{\mathbf{M}}_{0|t}$. Unfortunately, the $\hat{\mathbf{M}}_{0|t}$ predicted by VFDiff is usually inaccurate at the beginning of the reverse process (i.e. the value of $t$ is still relatively large). Therefore, in order to restore the binding energy gradient computed by VFNet as accurately as possible, we propose position-tuning. With the training of equation 4, VFNet has the ability to denoise the perturbed data, and the $\hat{\mathbf{M}}_{0|t}$ predicted by VFDiff can be regarded as a sample with fine noise added. Thus before calculating the posterior distribution of $\mathbf{M}_{t-1}$, we add $f_\psi(\hat{\mathbf{M}}_{0|t},\mathbf{P})$ to the value of $\{\hat{\mathbf{x}}_{0|t,i}\}_{i=1}^{N_{\mathcal{M}}}$ in $\hat{\mathbf{M}}_{0|t}$ to better approximate the ground truth $\mathbf{X}_0^{\mathcal{M}}$ and $\mathbf{F}_t$. Here, $f_\psi(\cdot,\cdot)$ is the output of the last layer of $\mathbf{x}^l$ in VFNet and it had been trained with equation 4.

**3D Equivariant Molecular Diffusion** We then apply a $L$ layers SE(3)-equivariant neural network on the $k$-nn graph of the protein-ligand complex (denoted as $\mathcal{C} = [\![\mathbf{M},\mathbf{P}]\!]$, where $[\![\cdot]\!]$ denotes concatenation along the first dimension) to learn the atom-wise protein-molecule interactions in generative process. The SE(3)-invariant atom embeddings $\mathbf{H}^{\mathcal{C}}$ and SE(3)-equivariant positions $\mathbf{X}^{\mathcal{C}}$ are updated as follows:

$$
\mathbf{h}_{t,i}^{\mathcal{C},l+1} = \mathbf{h}_{t,i}^{\mathcal{C},l} + \sum_{j\in\mathcal{N}_C(i),i\neq j} f_h^{\mathcal{C},l}(d_{ij}^l,\mathbf{h}_{t,i}^{\mathcal{C},l},\mathbf{h}_{t,j}^{\mathcal{C},l},\mathbf{e}_{ij})
$$
$$
\mathbf{x}_{t,i}^{\mathcal{C},l+1} = \mathbf{x}_{t,i}^{\mathcal{C},l} + \sum_{j\in\mathcal{N}_C(i),i\neq j} (\mathbf{x}_{t,i}^{\mathcal{C},l} - \mathbf{x}_{t,j}^{\mathcal{C},l}) f_x^{\mathcal{C},l}(d_{ij}^l,\mathbf{h}_{t,i}^{\mathcal{C},l+1},\mathbf{h}_{t,j}^{\mathcal{C},l+1},\mathbf{e}_{ij}) \cdot \mathbb{1}_{\text{mol}} \tag{12}
$$

where $\mathcal{N}_C(i)$ stands for the set of $k$-nearest neighbors of atom $i$ on the protein-ligand complex graph, $\mathbf{e}_{ij}$ indicates the atom $i$ and atom $j$ are both protein atoms, both ligand atoms, or one pro-

tein atom and one ligand atom, and $\mathbb{1}_{\text{mol}}$ is the ligand atom mask since the protein atom coordinates are known and thus supposed to remain unchanged during this update. We use $\hat{\mathbf{V}}_{0|1} = \text{softmax}(\text{MLP}(\mathbf{H}_{0,1:N_M}^{\mathcal{C},L}))$ and $\hat{\mathbf{X}}_{0|1} \bar{=} \mathbf{X}_{0,1:N_M}^{\mathcal{C},L}$ as the final prediction. We leave the details about the training and sampling procedures of VFDiff in Appendix E.5.

## 5 EXPERIMENTS

### 5.1 DATASETS

In order to accurately model the force field of a ligand molecule within a protein pocket, we used the molecule-protein pairs (complexes) binding affinity in the PDBbind v2016 dataset as a supervising signal to train VFNet. The PDBbind v2016 dataset consists of 3325 training complexes and 178 testing complexes, and it is commonly employed in binding-affinity prediction tasks. As for molecular generation, following the previous work Luo et al. (2021); Peng et al. (2022); Guan et al. (2023), we train and evaluate VFDiff on the CrossDocked2020 dataset Francoeur et al. (2020). We follow the same data preparation and splitting as Luo et al. (2021), where the 22.5 million docked binding complexes are refined to high-quality docking poses (RMSD between the docked pose and the ground truth $<1$Å) and diverse proteins (sequence identity $<30\%$). The resulting dataset consists of 100, 000 protein-ligand pairs for training and 100 proteins for testing. To ensure a fair comparison with our baseline methods, we adhere to identical data splits for training our guide models and evaluating our overall method.

### 5.2 BASELINE METHODS

We evaluate our method against state-of-the-art baselines in structure-based drug design (SBDD). These include liGAN (Ragoza et al. (2022)), which utilizes a conditional variational autoencoder (CVAE) trained on grid representations of atomic densities in protein-ligand structures. We also benchmark against autoregressive methods such as AR (Luo et al. (2021)) and Pocket2Mol (Peng et al. (2022)), which are GNN-based approaches that generate 3D molecular atoms by conditioning on the protein pocket and previously generated atoms. Moreover, our comparisons include recent diffusion-based methods like TargetDiff (Guan et al. (2023)) and DecompDiff (Guan et al. (2024)), which have established new benchmarks for the non-autoregressive generation of atom coordinates and types. DecompDiff improves upon TargetDiff by incorporating bond information and introducing decomposed priors for ligand arms and scaffolds. IRDiff (Huang et al. (2024a)) applies data retrieval augmented generation techniques from NLP to filter references from high-affinity libraries, boosting the affinity of generated molecules. IPDiff (Huang et al. (2024c)) is the current state-of-the-art diffusion method that integrates protein-ligand interactions, generating molecules more likely to fall within the desired property distributions.

### 5.3 METRICS

To evaluate the quality of molecules generated by VFDiff and the baselines, we adopt a multi-faceted assessment strategy encompassing molecular properties, conformation, and binding affinity with the target.

#### 5.3.1 MOLECULAR PROPERTIES

**QED (Quantitative Estimate of Druglikeness)** This metric evaluates a molecule's drug-likeness by reflecting the typical distribution of molecular properties in successful drug candidates (Bickerton et al. (2012)). **SA (Synthetic Accessibility)** SA assesses the ease with which a molecule can be synthesized, serving as a vital indicator of its practical manufacturability in a laboratory or industrial setting (Ertl & Schuffenhauer (2009); You et al. (2018)). **Diversity**. Diversity is measured as the average pairwise Tanimoto distances (Tanimoto (1958)) among all ligands generated for a specific protein pocket. **Bond Distance Distribution**. We calculate the Jensen Shannon divergences (JSD) to assess the differences in bond distance distributions between the reference molecules and the generated molecules (Lin (1991)).

### 5.3.2 BINDING AFFINITY

AutoDock Vina is employed to calculate the following metrics: **Vina Score**. Estimates binding affinity from the 3D pose of generated molecules, where a favorable score suggests strong binding potential. **Vina Min**. By conducting a local structure minimization prior to affinity estimation, this metric presents a slightly refined perspective on Vina Score. **Vina Dock**. Incorporating a re-docking process, Vina Dock showcases the optimal binding affinity that can be achieved. **High Affinity**. This is a comparative metric evaluating the percentage of generated molecules that manifest better binding than a reference molecule for a given protein pocket.

Figure 2: Comparing the distributions of all-atom distances for reference molecules in the test set (Gray) and model generated molecules (color). The Jensen-Shannon divergence (JSD) between the two distributions is reported.

Table 1: Jensen-Shannon Divergence comparing bond distance distributions between reference molecules and generated molecules. Lower values indicate better performance. '-' represents single bonds, '=' represents double bonds, and ':' represents aromatic bonds. The first and second-place results are emphasized with bold in red and underlined text in blue, respectively.

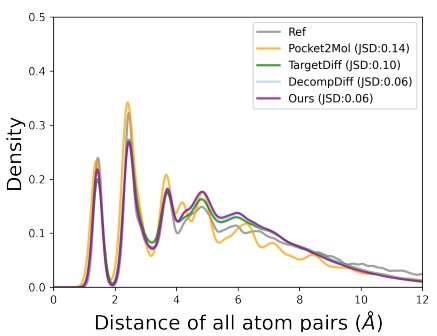

| Bond | liGAN | AR | Pocket2 Mol | Target Diff | Decomp Diff | IP Diff | IR Diff | VF Diff |
|------|-------|-------|-------|-------|-------|-------|-------|-------|
| C-C | 0.601 | 0.609 | 0.496 | 0.369 | 0.371 | 0.386 | 0.439 | **0.365** |
| C=C | 0.665 | 0.620 | 0.561 | 0.505 | 0.539 | 0.245 | 0.272 | **0.191** |
| C-N | 0.634 | 0.474 | 0.416 | 0.363 | 0.352 | 0.298 | 0.302 | **0.244** |
| C=N | 0.749 | 0.635 | 0.629 | 0.550 | 0.592 | 0.238 | 0.255 | **0.209** |
| C-O | 0.656 | 0.492 | 0.454 | 0.421 | 0.373 | 0.366 | 0.371 | **0.259** |
| C=O | 0.661 | 0.558 | 0.516 | 0.461 | 0.381 | **0.353** | 0.361 | 0.377 |
| C:C | 0.497 | 0.451 | 0.416 | 0.263 | 0.258 | 0.169 | 0.214 | **0.133** |
| C:N | 0.638 | 0.551 | 0.487 | 0.235 | 0.273 | **0.128** | 0.209 | 0.158 |

### 5.4 MAIN RESULTS

**Generated Molecular Structures** We compare the performance of our proposed method VFDiff against the above baseline methods. The all-atom pairwise distance distribution of the generated molecules are plotted in Figure 2. Table 1 presents the bond distributions of the molecules generated by different methods compared against the corresponding reference empirical distributions. Our VFDiff achieves superior performance on major bond types compared to all other methods, which demonstrating the ability of VFDiff in generating stable molecular structures.

**Target Binding Affinity and Molecule Properties** Figure 4 illustrates the mean Vina energy (computed by AutoDock Vina (Francoeur et al. (2020))) of all generated molecules for each binding pocket. Based on the Vina Score, the models with conditional guidance during generation (VFDiff and IPDiff ) have an absolute advantage, with 95% of the molecules with optimal Vina Score being generated by these two models. Among them, the conditional guidance introduced by VFDiff does not break the equivariant of the molecule, maintains the elegance of the diffusion and generation process, and outperforms IPDiff, which does not have an equivariant process, by as much as **22** percentage points.

As show in Table 2, our model outperforms existing state-of-the-art (SOTA) models in all docking metrics, with the most important evaluation metric, *Vina Score*, is substantially ahead of other baselines, achieving a $14.8\%$ improvement over the SOTA (IPDiff). Meanwhile, the difference between the re-docking score metric (*Vina Dock*) and the model-predicted docking attitude score (*Vina Score*) is optimal in our model among all diffusion-based models. This emphasize that our proposed method not only generates high-affinity molecules but also accurately reproduces docking positions in the pocket space.

In terms of molecular properties, VFDiff achieved first tier trade-off in both *QED* and *SA* metrics when comparing existing diffusion models. The auto-regressive model Pocket2Mol substantially outperforms the diffusion-based model in both *QED* and *SA* metrics. Future work on diffusion models should investigate the factors influencing the *SA* metric and explore ways to optimize it to approach the performance of auto-regressive models.

Table 2: Summary of binding affinity and molecular properties of reference molecules and molecules generated by VFDiff and baselines. (↑) / (↓) denotes whether a larger / smaller number is preferred. Top 2 results are bolded and underlined, respectively.

| Methods | Vina Score(↓) | | Vina Min (↓) | | Vina Dock (↓) | | High Affinity(↑) | | QED(↑) | | SA(↑) | | Diversity(↑) | |
|---|---|---|---|---|---|---|---|---|---|---|---|---|---|---|
| | Avg. | Med. | Avg. | Med. | Avg. | Med. | Avg. | Med. | Avg. | Med. | Avg. | Med. | Avg. | Med. |
| liGAN | - | - | - | - | -6.33 | -6.20 | 21.1% | 11.1% | 0.39 | 0.39 | 0.59 | 0.57 | 0.66 | 0.67 |
| GraphBP | - | - | - | - | -4.80 | -4.70 | 14.2% | 6.7% | 0.43 | 0.45 | 0.49 | 0.48 | **0.79** | **0.78** |
| AR | -5.75 | -5.64 | -6.18 | -5.88 | -6.75 | -6.62 | 37.9% | 31.0% | 0.51 | 0.50 | 0.63 | 0.63 | 0.70 | 0.70 |
| Pocket2Mol | -5.14 | -4.70 | -6.42 | -5.82 | -7.15 | -6.79 | 48.4% | 51.0% | **0.56** | **0.57** | **0.74** | **0.75** | 0.69 | 0.71 |
| TargetDiff | -5.47 | -6.30 | -6.64 | -6.83 | -7.80 | -7.91 | 58.1% | 59.1% | 0.48 | 0.48 | 0.58 | 0.58 | 0.72 | 0.71 |
| DecompDiff | -5.67 | -6.04 | -7.04 | -7.09 | -8.39 | -8.43 | 64.4% | 71.0% | 0.45 | 0.43 | 0.61 | 0.60 | 0.68 | 0.68 |
| IRDiff | -6.03 | -6.89 | -7.27 | -7.37 | -8.42 | -8.42 | 67.4% | 72.7% | 0.53 | 0.54 | 0.59 | 0.58 | 0.72 | 0.72 |
| IPDiff | -6.42 | -7.01 | -7.45 | -7.48 | -8.57 | -8.51 | 69.5% | 75.5% | 0.52 | 0.53 | 0.61 | 0.59 | 0.74 | 0.73 |
| **VFDiff** | **-7.37** | **-7.75** | **-8.18** | **-8.18** | **-8.77** | **-8.72** | **69.5%** | **75.5%** | 0.54 | 0.55 | 0.57 | 0.57 | 0.72 | 0.71 |
| Reference | -6.36 | -6.41 | -6.71 | -6.49 | -7.45 | -7.26 | - | - | 0.48 | 0.47 | 0.73 | 0.74 | - | - |

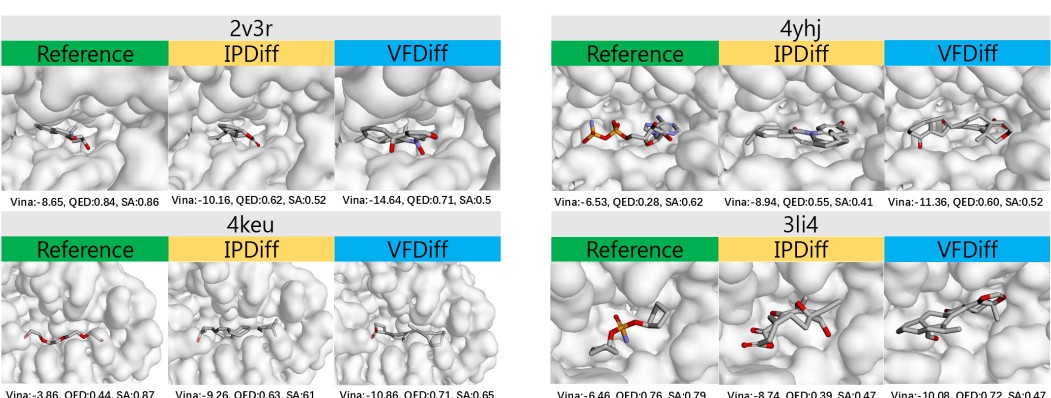

Figure 3: Examples of generated protein-pocket ligands (4yhj, 2v3r, 4keu and 3li4). Carbon atoms in reference ligands, ligands generated by IPDiff and VFDiff are visualized in green, yellow and blue, respectively. Vina Score, QED and SA are reported.

The chart clearly shows a negative correlation between the *Vina Dock* and High affinity metrics. Therefore, a model that performs well in *Vina Dock* will also excel in this metric. Our VFDiff outperformed all methods, achieving a performance of **69.5%**. In terms of metric diversity, our method also demonstrates the similar performance of exiting diffusion models.

## 5.5 MODEL ANALYSIS

**Influence of Energy-planning Force-guiding and Position-tuning** Structure-based molecule generation is aims to fulfill two key requirements: (1) the ligand molecules have to be spatially complementary to the protein pockets (2) they must match in terms of binding energy. We believe that traditional diffusion models only fit the spatial data distribution, resulting in generated molecules that perform poorly in binding energy, which is one of the most important metrics in SBDD. To solve the above problem, we propose energy-planning, force-guiding, and position-tuning mechanisms that use energy constraints to change the trajectories of atoms during forward and reverse processes, thus better modeling the data distribution. We showcase these methods efficacy in our VFDiff, and put results in Table 3.

We found that the affinity of the generated molecules is not enhanced by simply using the force-guiding strategy. This may be due to the inability of the standard diffusion model to accommodate this change in molecular conformational distribution at different moments. The above problem was solved when we trained the model with training data with energy offsets. The affinity, SA, and QED metrics are close to or even beyond the baseline level, but the results are still unsatisfactory. As we assumed before, even the baseline model with only position tuning performs very well. Because most of the current molecular generation and diffusion models use the $\mathbf{X}_0$-predicted strategy, which is a teach-forcing approach, the disadvantage is that the error generated when $\hat{\mathbf{X}}_0^{\mathcal{M}}$ is inac-

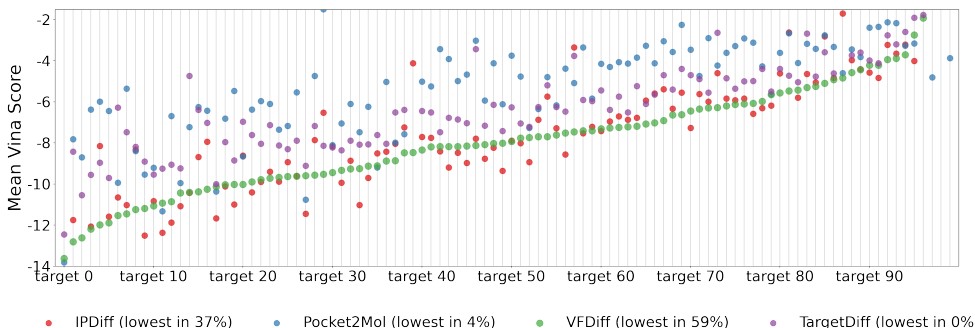

Figure 4: Mean Vina energy for different generated molecules (IPDiff vs. Pocket2Mol vs. VFDiff vs. TargetDiff) across 100 testing binding targets. Binding targets are sorted by the median Vina energy of VFDiff generated molecules. Lower Vina energy means a higher estimated binding affinity.

curately predicted may be amplified cumulatively over time, and the introduction of position-tuning minimizes this error as much as possible, which is good for our generation. To verify whether position-tuning improves the accuracy of molecular conformation prediction, we tested the trained VFNet and denoising diffusion model on the validation set. We first normalized the adjustment direction predicted by VFNet to a unit length and then scaled it by different scaling coefficients ($c$) before adding it to the diffusion-predicted output $\hat{\mathbf{X}}$. We compared the mean squared error between the predicted and ground-truth $\mathbf{X}$ at different time steps. As shown in the tabel below, VFNet significantly enhances the accuracy of molecular conformations, thereby validating our hypothesis. In

| (Scaling coefficient) | $c = 0$ | $c = 0.05$ | $c = 0.1$ | $c = 1$ | $c = 10$ |
|---|---|---|---|---|---|
| (Position loss) | 0.6995 | 0.6973 | 0.6876 | 0.7798 | 10.464 |

the end, we integrated all the methods mentioned above to obtain the VFDiff, which achieved the first tier in almost all the metrics. The Vina Score verifies the validity of our proposed energy path planning, and the diversity of the molecules also illustrates that such an approach does not trap the generated molecules in a local optimal solution.

Table 3: The effect of energy-planning, force-guiding and position-tuning mechanism. ($\uparrow$) / ($\downarrow$) denotes a larger / smaller number is better. The top 2 results are highlighted with bold and underlined text, respectively.

| Methods | Vina Score($\downarrow$) | | Vina Min ($\downarrow$) | | Vina Dock ($\downarrow$) | | High Affinity($\uparrow$) | | QED($\uparrow$) | | SA($\uparrow$) | | Diversity($\uparrow$) | |
|---|---|---|---|---|---|---|---|---|---|---|---|---|---|---|
| | Avg. | Med. | Avg. | Med. | Avg. | Med. | Avg. | Med. | Avg. | Med. | Avg. | Med. | Avg. | Med. |
| Baseline | -5.23 | -6.18 | -6.35 | -6.81 | -7.52 | -7.87 | 56.6% | 55.1% | 0.47 | 0.48 | 0.58 | 0.58 | 0.72 | 0.72 |
| + force-guiding | -4.50 | -5.36 | -5.77 | -5.79 | -7.13 | -7.16 | 49.6% | 46.4% | 0.41 | 0.40 | **0.61** | **0.60** | **0.80** | **0.79** |
| + energy-planning&force-guiding | -5.80 | -6.57 | -6.95 | -7.03 | -7.94 | -8.10 | 61.4% | 64.7% | 0.47 | 0.48 | 0.58 | 0.57 | 0.72 | 0.71 |
| + position-tuning | -6.93 | -7.24 | -7.70 | -7.68 | -8.26 | -8.21 | 68.1% | 74.2% | 0.54 | 0.53 | 0.56 | 0.57 | 0.69 | 0.68 |
| **VFDiff** | **-7.37** | **-7.75** | **-8.18** | **-8.18** | **-8.77** | **-8.72** | **69.5%** | **75.5%** | **0.54** | **0.55** | 0.57 | 0.57 | 0.72 | 0.71 |
| Reference | -6.36 | -6.41 | -6.71 | -6.49 | -7.45 | -7.26 | - | - | 0.48 | 0.47 | 0.73 | 0.74 | - | - |

# 6 CONCLUSION

The paper introduces VFDiff, an innovative SE(3)-equivariant diffusion model guided by vector fields for target-aware 3D molecule generation. The model represents a significant advancement in structure-based drug design (SBDD) by explicitly incorporating protein-ligand binding energy into the diffusion process. Unlike previous models that only focus on shape complementarity, VFDiff ensures both spatial and energetic compatibility between generated molecules and target protein pockets. Through mechanisms such as energy-planning, force-guiding, and position-tuning, the model achieves state-of-the-art performance on the CrossDocked2020 dataset, with superior binding affinity, molecular properties, and diversity. This novel approach bridges a critical gap in molecular generation by leveraging physical principles like force fields, offering both theoretical depth and practical applications in drug discovery.

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

# A APPENDIX

# B CONVENTIONAL TARGET-AWARE 3D MOLECULAR DIFFUSION PROCESS

In the forward process, a tiny Gaussian noise is continuously added to the original data so that the distribution of the data eventually takes on a standard normal distribution. The diffusion model generates new samples by learning the data distribution from this process. In previous work, this standard noise addition model was only applied to ligand molecules, and the transition kernel is shown below:

$$q(\mathbf{M}_t|\mathbf{M}_0, \mathbf{P}) = \prod_{i=1}^{N_M} \mathcal{N}(\mathbf{x}_{t,i}^{\mathcal{M}}; \sqrt{\bar{\alpha}_t}\mathbf{x}_0, (1-\bar{\alpha}_t)I) \cdot \mathcal{C}(\mathbf{v}_{t,i}^{\mathcal{M}}|\sqrt{\bar{\alpha}_t}\mathbf{v}_0 + (1-\bar{\alpha}_t)/K), \quad (13)$$

$$q(\mathbf{M}_t|\mathbf{M}_{t-1}, \mathbf{P}) = \prod_{i=1}^{N_M} \mathcal{N}(\mathbf{x}_{t,i}^{\mathcal{M}}; \sqrt{\alpha_t}(\mathbf{x}_{t-1}), (1-\alpha_t)I) \cdot \mathcal{C}(\mathbf{v}_{t,i}^{\mathcal{M}}|\sqrt{\alpha_t}\mathbf{v}_{t-1} + (1-\alpha_t)/K),$$
$$(14)$$

where $\mathcal{N}$ and $\mathcal{C}$ stand for the Gaussian and categorical distribution respectively, variance is defined by fixed variance schedules. The corresponding posterior can be analytically derived as follows:

$$q(\mathbf{M}_{t-1}|\mathbf{M}_t, \mathbf{M}_0, \mathbf{P}) = \prod_{i=1}^{N_M} \mathcal{N}\left(\mathbf{x}_{t-1,i}^{\mathcal{M}}; \widetilde{\boldsymbol{\mu}}(\mathbf{x}_{t,i}^{\mathcal{M}}, \mathbf{x}_{0,i}^{\mathcal{M}}), \widetilde{\beta}_t I\right) \cdot$$
$$\mathcal{C}\left(\mathbf{v}_{t-1,i}^{\mathcal{M}}|\boldsymbol{c}(\mathbf{v}_{t,i}^{\mathcal{M}}, \mathbf{v}_{0|t,i}^{\mathcal{M}})\right), \quad (15)$$

where $\widetilde{\boldsymbol{\mu}}(\mathbf{x}_{t,i}^{\mathcal{M}}, \mathbf{x}_{0,i}^{\mathcal{M}}) = \frac{\sqrt{\alpha_t}(1-\bar{\alpha}_{t-1})}{1-\bar{\alpha}_t}\mathbf{x}_t + \frac{\sqrt{\bar{\alpha}_{t-1}}(1-\alpha_t)}{1-\bar{\alpha}_t}\mathbf{x}_0 + \frac{\alpha_t(\bar{\alpha}_{t-1}-1)}{1-\bar{\alpha}_t}, \widetilde{\beta}_t = \frac{1-\bar{\alpha}_{t-1}}{1-\bar{\alpha}_t}\beta_t, \beta_t = 1-\alpha_t,$ $\bar{\alpha}_t = \prod_{s=1}^t \alpha_s, \boldsymbol{c}(\mathbf{v}_{t,i}^{\mathcal{M}}, \hat{\mathbf{v}}_{0|t,i}^{\mathcal{M}}) = \frac{\boldsymbol{c}^*}{\sum_{k=i}^{K} c_k^*}[\alpha\mathbf{v}_{t,i} + (1-\alpha_t)/K] \odot [\bar{\alpha}_{t-1}\mathbf{v}_{0,i} + (1-\bar{\alpha}_{t-1})/K].$

In the inverse generation process, we usually use the neural network parameterized by $\theta 1$ to fit the distribution at each moment, by approximating the initial moments $\hat{\mathbf{x}}_{0|t,i}^{\mathcal{M}}$ and $\hat{\mathbf{v}}_{0|t,i}^{\mathcal{M}}$, and solving by substituting Equation 15 as follows:

$$p_{\theta 1}(\mathbf{M}_{t-1}|\mathbf{M}_t, \mathbf{P}) = \prod_{i=1}^{N_M} \mathcal{N}\left(\mathbf{x}_{t-1,i}^{\mathcal{M}}; \widetilde{\boldsymbol{\mu}}(\mathbf{x}_{t,i}^{\mathcal{M}}, \hat{\mathbf{x}}_{0,i}^{\mathcal{M}}), \widetilde{\beta}_t I\right) \cdot$$
$$\mathcal{C}\left(\mathbf{v}_{t-1,i}^{\mathcal{M}}|\boldsymbol{c}(\mathbf{v}_{t,i}^{\mathcal{M}}, \hat{\mathbf{v}}_{0|t,i}^{\mathcal{M}})\right). \quad (16)$$

# C PROOFS

## C.1 DERIVATION OF FORWARD TRANSITION KERNELS OF VFDIFF

As mentioned in the introduction section, the goal of target-aware molecular generation is to generate molecular conformations that are spatially complementary and energetically matched to the target. Conventional diffusion models that model the diffusion process where the coordinate distribution of the molecules is actually weakly correlated with the given pockets, i.e., provide only the pocket structure of the center of mass information. We propose the VFDiff which attempts to incorporate information about the force field $\mathbf{F}$ to which the molecule is subjected in the pocket into the diffusion process, such that at each moment the binding energy of the molecule to the pocket is changing in a given direction.

Firstly, we have the marginal Gaussian for $\mathbf{X}_t$ and $\mathbf{X}_{t-1}$ as described in Equation 7:

$$q(\mathbf{X}_t|\mathbf{X}_0, \mathbf{P}, \mathbf{F}_t) = \mathcal{N}(\mathbf{X}_t; \sqrt{\bar{\alpha}_t}\mathbf{X}_0 + \sqrt{\alpha_t}\mathbf{S}_t, (1-\bar{\alpha}_t)\boldsymbol{\Sigma}),$$
$$q(\mathbf{X}_{t-1}|\mathbf{X}_0, \mathbf{P}, \mathbf{F}_{t-1}) = \mathcal{N}(\mathbf{X}_{t-1}; \sqrt{\bar{\alpha}_{t-1}}\mathbf{X}_0 + \sqrt{\alpha_{t-1}}\mathbf{S}_{t-1}, (1-\bar{\alpha}_{t-1})\boldsymbol{\Sigma}) \quad (17)$$
$$\mathbf{S}_t = \eta_3 \cdot k_t \cdot \mathbf{F}_t, \quad \mathbf{S}_{t-1} = \eta_3 \cdot k_{t-1} \cdot \mathbf{F}_{t-1},$$

we can assume that:

$$q(\mathbf{X}_t|\mathbf{X}_{t-1}, \mathbf{X}_0, \mathbf{P}, \mathbf{F}_t) = \mathcal{N}(\mathbf{X}_t; \mathbf{A}(\mathbf{X}_{t-1} + \mathbf{b}), \mathbf{L}^{-1}) \tag{18}$$

and we can derive the marginal Gaussian for $\mathbf{X}_t$ according to equations 17 and 18, for all $t > 1$:

$$q(\mathbf{X}_t|\mathbf{X}_0, \mathbf{P}, \mathbf{F}_t) = \mathcal{N}(\mathbf{X}_t; \mathbf{A}(\sqrt{\bar{\alpha}_{t-1}}\mathbf{X}_0 + \sqrt{\alpha_{t-1}}\mathbf{S}_{t-1} + \mathbf{b}), \mathbf{L}^{-1} + (1 - \bar{\alpha}_{t-1})\mathbf{A}\boldsymbol{\Sigma}\mathbf{A}^{\mathrm{T}}), \tag{19}$$

therefore, we can derive that:

$$\begin{aligned}
\mathbf{A} &= \sqrt{\alpha_t}\mathbf{I} \\
\mathbf{b} &= \mathbf{S}_t - \sqrt{\alpha_{t-1}}\mathbf{S}_{t-1} \\
\mathbf{L}^{-1} &= (1 - \alpha_t)\boldsymbol{\Sigma}
\end{aligned} \tag{20}$$

Particularly, according to Equations 17 and 18, we have:

$$\begin{aligned}
q(\mathbf{X}_1|\mathbf{X}_0, \mathbf{P}, \mathbf{F}_1) &= \mathcal{N}(\mathbf{X}_1; \sqrt{\bar{\alpha}_1}\mathbf{X}_0 + \sqrt{\alpha_1}\mathbf{S}_1, (1 - \bar{\alpha}_1)\boldsymbol{\Sigma}) \\
&= \mathcal{N}(\mathbf{X}_1; \sqrt{\alpha_1}(\mathbf{X}_0 + \mathbf{S}_1 - \sqrt{\alpha_0}\mathbf{S}_0), (1 - \alpha_1)\boldsymbol{\Sigma})
\end{aligned} \tag{21}$$

For making Equation 21 to be hold, we set $\alpha_0 = 1$ and $\mathbf{S}_0 = \mathbf{O}$.

## C.2 DERIVATION OF THE POSTERIOR DISTRIBUTION OF THE SHIFTED DIFFUSION PROCESS

Following Luo et al. (2021), For all $t > 1$, according to the Bayes' rule:

$$q(\mathbf{X}_{t-1}|\mathbf{X}_t, \mathbf{X}_0, \mathbf{P}, \mathbf{F}) \quad (\mathbf{F} \text{ contains information about } \mathbf{F}_t \text{ and } \mathbf{F}_{t-1}) \tag{22}$$

$$= \frac{q(\mathbf{X}_t|\mathbf{X}_{t-1}, \mathbf{X}_0, \mathbf{P}, \mathbf{F})q(\mathbf{X}_{t-1}|\mathbf{X}_0, \mathbf{P}, \mathbf{F})}{q(\mathbf{X}_t|\mathbf{X}_0, \mathbf{P}, \mathbf{F})} \tag{23}$$

$$= \frac{\mathcal{N}(\mathbf{X}_t; \sqrt{\alpha_t}(\mathbf{X}_{t-1} + \mathbf{S}_t - \sqrt{\alpha_t}\mathbf{S}_{t-1}), \beta_t\boldsymbol{\Sigma}) \cdot \mathcal{N}(\mathbf{X}_{t-1}; \sqrt{\bar{\alpha}_{t-1}}\mathbf{X}_0 + \sqrt{\alpha_{t-1}}\mathbf{S}_{t-1}, \bar{\beta}_{t-1}\boldsymbol{\Sigma})}{\mathcal{N}(\mathbf{X}_t; \sqrt{\bar{\alpha}_t}\mathbf{X}_0 + \sqrt{\alpha_t}\mathbf{S}_t, (1 - \bar{\alpha}_t)\boldsymbol{\Sigma})} \tag{24}$$

$$\propto \exp\left\{-\frac{1}{2}\left[\frac{(\mathbf{X}_t - \sqrt{\alpha_t}(\mathbf{X}_{t-1} + \mathbf{S}_t - \sqrt{\alpha_t}\mathbf{S}_{t-1}))^2}{(1 - \alpha_t)} + \frac{(\mathbf{X}_{t-1} - \sqrt{\bar{\alpha}_{t-1}}\mathbf{X}_0 - \sqrt{\alpha_{t-1}}\mathbf{S}_{t-1})^2}{(1 - \bar{\alpha}_{t-1})}\right.\right. \tag{25}$$

$$\left.\left. - \frac{(\mathbf{X}_t - \sqrt{\bar{\alpha}_t}\mathbf{X}_0 - \sqrt{\alpha_t}\mathbf{S}_t)^2}{(1 - \bar{\alpha}_t)}\right]\right\}$$

$$= \exp\left\{-\frac{1}{2}\left[\frac{\alpha_t\mathbf{X}_{t-1}^2 + 2\alpha_t\mathbf{S}_t\mathbf{X}_{t-1} - 2\alpha_t\sqrt{\alpha_{t-1}}\mathbf{S}_{t-1}\mathbf{X}_{t-1} - 2\sqrt{\alpha_t}\mathbf{X}_t\mathbf{X}_{t-1}}{1 - \alpha_t}\right.\right. \tag{26}$$

$$\left.\left. + \frac{\mathbf{X}_{t-1}^2 - 2\sqrt{\bar{\alpha}_{t-1}}\mathbf{X}_{t-1}\mathbf{X}_0 - 2\sqrt{\alpha_t}\mathbf{S}_{t-1}\mathbf{X}_{t-1}}{1 - \bar{\alpha}_{t-1}}\right]\right\} + C(\mathbf{X}_0, \mathbf{X}_t)$$

$$\propto \exp\left\{-\frac{1}{2}\left[(\frac{\alpha_t}{1 - \alpha_t} + \frac{1}{1 - \bar{\alpha}_{t-1}})\mathbf{X}_{t-1}^2 + 2(\frac{-\sqrt{\alpha_t}\mathbf{X}_t + \alpha_t\mathbf{S}_t - \alpha_t\sqrt{\alpha_{t-1}}\mathbf{S}_{t-1}}{1 - \alpha_t}\right.\right. \tag{27}$$

$$\left.\left. - \frac{\sqrt{\bar{\alpha}_{t-1}}\mathbf{X}_0 + \sqrt{\alpha_{t-1}}\mathbf{S}_{t-1}}{1 - \bar{\alpha}_{t-1}})\mathbf{X}_{t-1}\right]\right\}$$

$$= \exp\left\{-\frac{1}{2}(\frac{1 - \bar{\alpha}_t}{(1 - \alpha_t)(1 - \bar{\alpha}_{t-1})})[\mathbf{X}_{t-1}^2\right.$$

$$\left. + 2(\frac{\mathbf{S}_t\alpha_t - \sqrt{\alpha_t}\mathbf{X}_t - \bar{\alpha}_t\mathbf{S}_t + \bar{\alpha}_t\sqrt{\alpha_{t-1}}\mathbf{S}_{t-1} + \sqrt{\alpha_t}\bar{\alpha}_{t-1}\mathbf{X}_t - (\sqrt{\bar{\alpha}_{t-1}}\mathbf{X}_0 + \sqrt{\alpha_{t-1}}\mathbf{S}_{t-1} - \alpha_t\sqrt{\bar{\alpha}_{t-1}}\mathbf{X}_0)}{1 - \bar{\alpha}_t})\mathbf{X}_{t-1}]\right\} \tag{28}$$

$$= \exp\left\{-\frac{1}{2}(\frac{1}{\frac{1 - \bar{\alpha}_t}{(1 - \alpha_t)(1 - \bar{\alpha}_{t-1})}})[\mathbf{X}_{t-1}^2\right.$$

$$\left. - 2(\frac{\sqrt{\bar{\alpha}_{t-1}}\beta_t}{1 - \bar{\alpha}_t}\mathbf{X}_0 + \frac{\sqrt{\alpha_t}(1 - \bar{\alpha}_{t-1})}{1 - \bar{\alpha}_t}\mathbf{X}_t - \frac{\alpha_t(1 - \bar{\alpha}_{t-1})}{1 - \bar{\alpha}_t}\mathbf{S}_t + \mathbf{S}_{t-1})\mathbf{X}_{t-1}]\right\} \tag{29}$$

$$\propto \mathcal{N}(\mathbf{X}_{t-1}; \frac{\sqrt{\alpha_t}(1 - \bar{\alpha}_{t-1})}{1 - \bar{\alpha}_t}\mathbf{X}_t + \frac{\sqrt{\bar{\alpha}_{t-1}}(1 - \alpha_t)}{1 - \bar{\alpha}_t}\mathbf{X}_0 + \frac{\alpha_t(\bar{\alpha}_{t-1} - 1)}{1 - \bar{\alpha}_t}\mathbf{S}_t + \sqrt{\alpha_{t-1}}\mathbf{S}_{t-1}, \frac{(1 - \bar{\alpha}_{t-1})}{1 - \bar{\alpha}_t}\beta_t\boldsymbol{\Sigma}) \tag{30}$$

Observing Equation 30, it can be found that to solve for the coordinate distribution of the molecule at moment $t$ at moment $t-1$ we need to know $\mathbf{X}_t^{\mathcal{M}}$ and $\mathbf{X}_{t-1}^{\mathcal{M}}$ as well as $\mathbf{S}_t$ and $\mathbf{S}_{t-1}$, where the value of $\mathbf{M}_0 = [\mathbf{X}_0^{\mathcal{M}}, \mathbf{V}_0^{\mathcal{M}}]$ at moment t is known, and as shown in Equations (1,2,3,5), $\mathbf{S}$ is a function of $\mathbf{P}$ and $\mathbf{M}$. Therefore, we only need a neural network $\phi_{\theta 1}(\mathbf{X}_t, t)$ to predict $\hat{\mathbf{M}}_{0|t} = [\hat{\mathbf{X}}_{0|t}^{\mathcal{M}}, \hat{\mathbf{V}}_{0|t}^{\mathcal{M}}]$, which can be substituted into the above equation to solve for the molecular conformation at moment $t-1$:

$$
\begin{aligned}
p_{\theta 1}(\mathbf{X}_{t-1}|\mathbf{X}_t, \mathbf{P}, \mathbf{F}) =&\mathcal{N}(\mathbf{X}_{t-1}; \frac{\sqrt{\alpha_t}(1-\bar{\alpha}_{t-1})}{1-\bar{\alpha}_t}\mathbf{X}_t + \frac{\sqrt{\bar{\alpha}_{t-1}}(1-\alpha_t)}{1-\bar{\alpha}_t}\phi_{\theta 1}(\mathbf{X}_t, t)+ \\
&\frac{\alpha_t(\bar{\alpha}_{t-1}-1)}{1-\bar{\alpha}_t}\hat{\mathbf{S}}_t + \sqrt{\alpha_{t-1}}\hat{\mathbf{S}}_{t-1}, \frac{(1-\bar{\alpha}_{t-1})}{1-\bar{\alpha}_t}\beta_t\mathbf{\Sigma})
\end{aligned}
\tag{31}
$$

### C.3 Derivation of the training objectives

Following , the training objectives can be represented as:

$$
\begin{aligned}
L =&\mathbb{E}_q\{-\log p_{\theta 1}(\mathbf{X}_0|\mathbf{X}_1, \mathbf{P}, \mathbf{F}) + \mathcal{D}_{KL}[q(\mathbf{X}_T|\mathbf{X}_0, \mathbf{P}, \mathbf{F})\|p_{\theta 1}(\mathbf{X}_T)] \\
&\sum_{t=2}^T \mathcal{D}_{KL}[q(\mathbf{X}_{t-1}|\mathbf{X}_t, \mathbf{X}_0, \mathbf{P}, \mathbf{F})\|p_{\theta 1}(\mathbf{X}_{t-1}|\mathbf{X}_t, \mathbf{P}, \mathbf{F})]\}
\end{aligned}
\tag{32}
$$

For the first and the second terms, we can derive them as constants c and discard them in the objective function. For the third term, we can derive it by Gaussian Keullback-Leibler divergence:

$$
\begin{aligned}
&\mathcal{D}_{KL}[q(\mathbf{X}_{t-1}|\mathbf{X}_t, \mathbf{X}_0, \mathbf{P}, \mathbf{F})\|p_{\theta 1}(\mathbf{X}_{t-1}|\mathbf{X}_t, \mathbf{P}, \mathbf{F})] \\
&= \frac{1}{2}\|\frac{\sqrt{\bar{\alpha}_{t-1}}(1-\alpha_t)}{1-\bar{\alpha}_t}(\phi_{\theta 1}(\mathbf{X}_t, t) - \mathbf{X}_0) + \frac{\alpha_t(\bar{\alpha}_{t-1}-1)}{1-\bar{\alpha}_t}(\hat{\mathbf{S}}_t - \mathbf{S}_t) + \sqrt{\alpha_{t-1}}(\hat{\mathbf{S}}_{t-1} - \mathbf{S}_{t-1})\|^2_{(\frac{(1-\bar{\alpha}_{t-1})}{1-\bar{\alpha}_t}\beta_t\mathbf{\Sigma})^{-1}}
\end{aligned}
\tag{33}
$$

Assuming that $\mathbf{S}_t$ is Lipschitz continuous w.r.t $\mathbf{X}_0$, then we can simplify the Gaussian Keullback-Leibler divergence:

$$
\begin{aligned}
&\frac{1}{2}\|\frac{\sqrt{\bar{\alpha}_{t-1}}(1-\alpha_t)}{1-\bar{\alpha}_t}(\phi_{\theta 1}(\mathbf{X}_t, t) - \mathbf{X}_0) + \frac{\alpha_t(\bar{\alpha}_{t-1}-1)}{1-\bar{\alpha}_t}(\hat{\mathbf{S}}_t - \mathbf{S}_t) + \sqrt{\alpha_{t-1}}(\hat{\mathbf{S}}_{t-1} - \mathbf{S}_{t-1})\|^2_{(\frac{(1-\bar{\alpha}_{t-1})}{1-\bar{\alpha}_t}\beta_t\mathbf{\Sigma})^{-1}} \\
&\le \mathbf{c}_t\|(\phi_{\theta 1}(\mathbf{X}_t, t) - \mathbf{X}_0) + (\hat{\mathbf{S}}_t - \mathbf{S}_t) + \sqrt{\alpha_{t-1}}(\hat{\mathbf{S}}_{t-1} - \mathbf{S}_{t-1})\| \\
&\le \mathbf{c}_t\|(\phi_{\theta 1}(\mathbf{X}_t, t) - \mathbf{X}_0)\| + \|(\hat{\mathbf{S}}_t - \mathbf{S}_t)\| + \|\sqrt{\alpha_{t-1}}(\hat{\mathbf{S}}_{t-1} - \mathbf{S}_{t-1})\| \\
&\le \gamma_t\|(\phi_{\theta 1}(\mathbf{X}_t, t) - \mathbf{X}_0)\|,
\end{aligned}
\tag{34}
$$

by the lipschitz continuity of Mahalanobis Distances and $\mathbf{S}_t$. Here $\mathbf{c}_t, \gamma_t$ are scaling factors. Finally, the training objective of atom position at time step $T = t-1$ are defined as follows:

$$
L_{t-1}^{(x)} = \frac{1}{2\tilde{\beta}_t^2}\sum_{i=1}^{N_M}\|\widetilde{\boldsymbol{\mu}}(\mathbf{x}_{t,i}^{\mathcal{M}}, \mathbf{x}_{0,i}^{\mathcal{M}}, \mathbf{f}_i^{\mathcal{M}}) - \widetilde{\boldsymbol{\mu}}(\mathbf{x}_{t,i}^{\mathcal{M}}, \hat{\mathbf{x}}_{0,i}^{\mathcal{M}}, \mathbf{f}_i^{\mathcal{M}})\|^2 = \gamma_t\sum_{i=1}^{N_M}\|\mathbf{x}_{0,i}^{\mathcal{M}} - \hat{\mathbf{x}}_{0,i}^{\mathcal{M}}\|;
$$

(35)

where $\hat{\mathbf{X}}_0$ and $\hat{\mathbf{V}}_0$ are predicted from $\mathbf{X}_t$ and $\mathbf{V}_t$, where $\gamma_t$ is a scaling factor. And we use the same objective function of atom type at time step $t-1$ as Guan et al. (2023)

### C.4 SE(3)-Invariant probability density function

**Zero Center of Mass (CoM)** It has been shown that the invariance to translational and rotational transformations is an important factor for the success of 3D molecule modeling. Therefore, we can use the element with zero center of mass (CoM) to represent the original atom coordinates, We will place the original element at $CoM = 0$, *i.e.*, $\frac{1}{N^{\mathcal{P}}}\sum_{i=1}^{N^{\mathcal{P}}}\mathbf{x}^i = \mathbf{0}$. We act on the CoM at the initial moment $t = 0$ in the forward process and at the moment $t = T$ in the reverse process to ensure

that the probability density function (PDF) at each moment of the entire Markov process is SE(3)-Equavariant. The following is our proof:

Because of the uniqueness of the center of mass, the translation transformation will have no effect on the coordinates of element, and thus the PDF is invariant to the translation transformation. Therefore it is only necessary to show that the PDF is SO(3)-Invariant. Kindly recall that $q(\mathbf{X}_t|\mathbf{X}_0, \mathbf{P}, \mathbf{F}_t) = \prod_{i=1}^{N_M} \mathcal{N}(\mathbf{x}_{t,i}^{\mathcal{M}}; \sqrt{\bar{\alpha}_t}\mathbf{x}_0 + \sqrt{\alpha_t}\mathbf{s}_{t,i}^{\mathcal{M}}, (1 - \bar{\alpha}_t)I)$. It follows from Equations 2 and 3 that $\mathbf{F}_t$ is a function of $\mathbf{X}_0^{\mathcal{M}}, \mathbf{V}_0^{\mathcal{M}}$ and $\mathbf{P}$. Thus when the rotation matrix $\mathbf{R} \in \mathbb{R}^{3\times3}$ acts on $\mathbf{X}_0^{\mathcal{M}}$ and $\mathbf{P}$, the value of $\mathbf{F}_t$ changes. Next we will show that this change is SE(3)-equivariant. Observe the following equation:

$$
\begin{aligned}
y &= \sum_{i \in N_{\mathcal{M}}} \left\| \mathbf{x}_i^L - \mathbf{x}_i^0 \right\|_2^2 \\
\mathbf{F} &= -\nabla_{\mathbf{X}^{\mathcal{M}}} y,
\end{aligned}
\tag{36}
$$

where $f_\psi(\mathbf{x}_t, \mathbf{V}_0^{\mathcal{M}}, \mathbf{P}) = \mathbf{x}^L$ is the output of the last layer of the VFNet with respect to $\mathbf{x}$, and the VFNet is an equivariant graph neural network (EGNN) Garcia Satorras et al. (2021), hence $\mathbf{x}^L$ is SE(3)-equivariant under the CoM system. It is easy to see that $y = \sum_{i \in N_{\mathcal{M}}} \left\| \mathbf{x}_i^L - \mathbf{x}_i^0 \right\|_2^2$ keeps non-change with the rotation transformation. For simplicity, we let $y = g_\theta(\mathbf{X}_t^{\mathcal{M}}, \mathbf{V}_0^{\mathcal{M}}, \mathbf{P})$, $R(\mathbf{X}) = \mathbf{R}\mathbf{X}$ and based on the previous conclusion we have the following equation:

$$
\begin{aligned}
p(\mathbf{X}_t^{\mathcal{M}}|\mathbf{X}_0^{\mathcal{M}}) &= p(\mathbf{R}\mathbf{X}_t^{\mathcal{M}}|\mathbf{R}\mathbf{X}_0^{\mathcal{M}}) \\
y &= g_\theta(\tilde{\mathbf{X}}_t, \mathbf{P}) \\
\mathbf{F} &= -\nabla_{\tilde{\mathbf{X}}_t^{\mathcal{M}}} g_\theta(\tilde{\mathbf{X}}_t^{\mathcal{M}}, \mathbf{V}_0^{\mathcal{M}}, \mathbf{P}) \\
\mathbf{F}' &= -\nabla_{\mathbf{R}\tilde{\mathbf{X}}_t^{\mathcal{M}}} g_\theta(\mathbf{R}\tilde{\mathbf{X}}_t^{\mathcal{M}}, \mathbf{V}_0^{\mathcal{M}}, \mathbf{R}\mathbf{P}),
\end{aligned}
\tag{37}
$$

where $p(\mathbf{X}_t^{\mathcal{M}}|\mathbf{X}_0^{\mathcal{M}})$ defined in equation 13 is a SE(3)-equivariant transformation kernel in CoM system. According to the chain rule we have:

$$
\begin{aligned}
-\nabla_{\tilde{\mathbf{X}}_t^{\mathcal{M}}} g_\theta(\mathbf{R}\tilde{\mathbf{X}}_t^{\mathcal{M}}, \mathbf{V}_0^{\mathcal{M}}, \mathbf{R}\mathbf{P}) &= -\frac{\partial g}{\partial R} \nabla_{\mathbf{R}\tilde{\mathbf{X}}_t^{\mathcal{M}}} g_\theta(\mathbf{R}\tilde{\mathbf{X}}_t^{\mathcal{M}}, \mathbf{V}_0^{\mathcal{M}}, \mathbf{R}\mathbf{P}) \\
&= \mathbf{R}^T \cdot \mathbf{F}'
\end{aligned}
\tag{38}
$$

Recall that $y = \sum_{i \in N_{\mathcal{M}}} \left\| \mathbf{x}_i^L - \mathbf{x}_i^0 \right\|_2^2$ is SE(3)-invariant, thus

$$
\begin{aligned}
y &= g_\theta(\tilde{\mathbf{X}}_t^{\mathcal{M}}, \mathbf{V}_0^{\mathcal{M}}, \mathbf{P}) \\
&= g_\theta(\mathbf{R}\tilde{\mathbf{X}}_t^{\mathcal{M}}, \mathbf{V}_0^{\mathcal{M}}, \mathbf{R}\mathbf{P}).
\end{aligned}
\tag{39}
$$

Bringing the above equation into Equation 38:

$$
\begin{aligned}
-\nabla_{\tilde{\mathbf{X}}_t^{\mathcal{M}}} g_\theta(\mathbf{R}\tilde{\mathbf{X}}_t, \mathbf{V}_0^{\mathcal{M}}, \mathbf{R}\mathbf{P}) &= -\nabla_{\tilde{\mathbf{X}}_t^{\mathcal{M}}} g_\theta(\tilde{\mathbf{X}}_t, \mathbf{V}_0^{\mathcal{M}}, \mathbf{P}) \\
&= \mathbf{F},
\end{aligned}
\tag{40}
$$

we have

$$
\begin{aligned}
\mathbf{F} &= \mathbf{R}^T \cdot \mathbf{F}' \\
\mathbf{R}\mathbf{F} &= \mathbf{R}\mathbf{R}^T \cdot \mathbf{F}' \\
\mathbf{R}\mathbf{F} &= \mathbf{F}'.
\end{aligned}
\tag{41}
$$

Thus $\mathbf{s}_i^{\mathcal{M}}$ in Equation 5 is also SE(3)-equivariant and we can derived that:

$$
q(\mathbf{R}\mathbf{X}_t^{\mathcal{M}}|\mathbf{R}\mathbf{X}_0^{\mathcal{M}}, \mathbf{R}\mathbf{P}, \mathbf{F}')
$$

$$
= \prod_{i=1}^{N_M} \mathcal{N}(\mathbf{x}_{t,i}^{\mathcal{M}}; \sqrt{\bar{\alpha}_t}\mathbf{R}\mathbf{x}_0^{\mathcal{M}} + \sqrt{\alpha_t}\mathbf{R}\mathbf{s}_{t,i}^{\mathcal{M}}, (1-\bar{\alpha}_t)\boldsymbol{\Sigma})
$$

$$
= \prod_{i=1}^{N_M} \frac{1}{(2\pi)^{3/2}\sqrt{(1-\bar{\alpha}_t)\boldsymbol{\Sigma}}}
$$

$$
\exp\{\frac{(\mathbf{R}\mathbf{x}_{t,i}^{\mathcal{M}} - \mathbf{R}\mathbf{x}_0^{\mathcal{M}} - \sqrt{\alpha_t}\mathbf{R}\mathbf{s}_{t,i}^{\mathcal{M}})^T((1-\bar{\alpha}_t)\boldsymbol{\Sigma})^{-1}(\mathbf{R}\mathbf{x}_{t,i}^{\mathcal{M}} - \mathbf{R}\mathbf{x}_0^{\mathcal{M}} - \sqrt{\alpha_t}\mathbf{R}\mathbf{s}_{t,i}^{\mathcal{M}})}{2}\}
$$

$$
= \prod_{i=1}^{N_M} \frac{1}{(2\pi)^{3/2}\sqrt{(1-\bar{\alpha}_t)\boldsymbol{\Sigma}}}
$$

$$
\exp\{\frac{(\mathbf{x}_{t,i}^{\mathcal{M}} - \mathbf{x}_0^{\mathcal{M}} - \sqrt{\alpha_t}\mathbf{s}_{t,i}^{\mathcal{M}})^T\mathbf{R}^T(1-\bar{\alpha}_t)\mathbf{R}(\mathbf{x}_{t,i}^{\mathcal{M}} - \mathbf{x}_0^{\mathcal{M}} - \sqrt{\alpha_t}\mathbf{s}_{t,i}^{\mathcal{M}})}{2}\} \tag{42}
$$

$$
= \prod_{i=1}^{N_M} \frac{1}{(2\pi)^{3/2}\sqrt{(1-\bar{\alpha}_t)\boldsymbol{\Sigma}}}
$$

$$
\exp\{\frac{1}{(1-\bar{\alpha}_t)}\frac{(\mathbf{x}_{t,i}^{\mathcal{M}} - \mathbf{x}_0^{\mathcal{M}} - \sqrt{\alpha_t}\mathbf{s}_{t,i}^{\mathcal{M}})^T(\mathbf{x}_{t,i}^{\mathcal{M}} - \mathbf{x}_0^{\mathcal{M}} - \sqrt{\alpha_t}\mathbf{s}_{t,i}^{\mathcal{M}})}{2}\}
$$

$$
= \prod_{i=1}^{N_M} \frac{1}{(2\pi)^{3/2}\sqrt{(1-\bar{\alpha}_t)\boldsymbol{\Sigma}}}
$$

$$
\exp\{\frac{(\mathbf{x}_{t,i}^{\mathcal{M}} - \mathbf{x}_0^{\mathcal{M}} - \sqrt{\alpha_t}\mathbf{s}_{t,i}^{\mathcal{M}})^T((1-\bar{\alpha}_t)\boldsymbol{\Sigma})^{-1}(\mathbf{x}_{t,i}^{\mathcal{M}} - \mathbf{x}_0^{\mathcal{M}} - \sqrt{\alpha_t}\mathbf{s}_{t,i}^{\mathcal{M}})}{2}\}
$$

$$
= q(\mathbf{X}_t^{\mathcal{M}}|\mathbf{X}_0^{\mathcal{M}}, \mathbf{P}, \mathbf{F}).
$$

Now, we finish our proof.

## D  More Ablation Study

### D.1  Effect of the Shifting Scales

As mentioned in Equation 5, the shift $\mathbf{S}_t$ at the time step $t$ consists of a coefficient $k_t$ and a 3-dim vector field generated by VFNet $\psi_\theta(\cdot)$, where the coefficient $k_t = \sqrt{\bar{\alpha}} \cdot (1 - \sqrt{\bar{\alpha}})$ and $\eta_3$ is a hyper-parameter to adjust the scale of the shifts in the diffusion trajectory. To inverstigate the effect of different shifting scales, we set the $\eta_3$ to 4 values: (1) $\eta_3 = 0$, (2) $\eta_3 = 1$, (3) $\eta_3 = 10$, (4) $\eta_3 = 20$ and present the results in the Table 5. It worth noting that $\eta_3 = 0$ indicates the energy-planning and force-guiding mechanisms are removed from VFDiff. We find that when $\eta_3$ is small (i.e. $\eta_3 = 1$) the binding energy of the generated molecules is comparable to that at $\eta_3 = 0$. The reason for this may be that the trajectories of the molecules are roughly the same as before and therefore the model fails to explore the energy landscape. when $\eta_3 = 10$, the trajectories of the molecules change more significantly than before, and the model is able to find a pattern that allows for the generation of molecules with good binding energies. However, at $\eta_3 = 20$, the performance decreases, probably due to the fact that the given shifetd bias is too large and the trajectory is no longer traceable, thus affecting the final result.

### D.2  Effect of the Adjusted Labels

As mentioned in Equation 2, we multiply the binding affinity label $y$ by a scaling factor in order to give the model a better sense of how the binding energy is changing, and thus better guide the evolutionary trajectory of the molecule. In order to verify our suspicion, we tested the effect of VFNet in real molecule generation without and with the scaling factor added, and the results are shown in Table 4.

Table 4: The effect of the adjusted labels (al) on binding-related metrics. ($\uparrow$) / ($\downarrow$) denotes a larger / smaller number is better. Top 1 result is highlighted with **bold text**.

| Methods | Vina Score ($\downarrow$) | | Vina Min ($\downarrow$) | | Vina Dock ($\downarrow$) | | High Affinity ($\uparrow$) | |
| --- | --- | --- | --- | --- | --- | --- | --- | --- |
| | Avg. | Med. | Avg. | Med. | Avg. | Med. | Avg. | Med. |
| VFDiff | **-7.37** | **-7.75** | **-8.18** | **-8.18** | **-8.77** | **-8.72** | **69.5%** | **75.5%** |
| VFDiff w/o al | -7.13 | -7.46 | -8.02 | -8.08 | -8.58 | -8.61 | 69.2% | 75.1% |

Table 5: The effect of the different shifting scales on binding-related metrics. ($\uparrow$) / ($\downarrow$) denotes a larger / smaller number is better. Top 1 result is highlighted with **bold text**.

| Methods | Vina Score ($\downarrow$) | | Vina Min ($\downarrow$) | | Vina Dock ($\downarrow$) | | High Affinity ($\uparrow$) | |
| --- | --- | --- | --- | --- | --- | --- | --- | --- |
| | Avg. | Med. | Avg. | Med. | Avg. | Med. | Avg. | Med. |
| Baseline | -5.23 | -6.18 | -6.35 | -6.81 | -7.52 | -7.87 | 56.6% | 55.1% |
| $\eta_3 = 0$ | -6.93 | -7.24 | -7.70 | -7.68 | -8.26 | -8.21 | 68.1% | 74.2% |
| $\eta_3 = 1$ | -6.99 | -7.33 | -7.79 | -7.76 | -8.32 | -8.38 | 68.9% | 74.6% |
| $\eta_3 = 10$ | **-7.37** | **-7.75** | **-8.18** | **-8.18** | -8.77 | **-8.72** | **69.5%** | **75.5%** |
| $\eta_3 = 20$ | -7.21 | -7.64 | -8.15 | -8.17 | **-8.81** | -8.71 | 69.3% | 75.4% |

# E    IMPLEMENTATION DETAILS

## E.1    INITIALIZATION OF INPUT

Following Guan et al. (2023), we use a one-hot element indicator {H, C, N, O, S, Se} and one-hot amino acid type indicator (20 types) to represent each protein atom. Similarly, each ligand atom are repsented with a one-hot element indicator {C, N, O, F, P, S, Cl}. And an additional one-dimensional flag indicating whether the atoms belong to the protein or ligand are introduced. Two linear layer are used to map the input protein and ligand into 128-dim latent spaces respectively.

## E.2    VFNET TRAINING

During the training, we use the Mean Squared Error (MSE) loss with respect to the difference between the predicted and ground truth binding affinity scores as the optimization objective. The binding affinity values of protein-ligand pairs range from 2.0 to 11.92. For avoiding information leakage, we filter the training set by calculating the Tanimoto similarity with the molecules in the testing set of CrossDocked2020, and the similarity threshold was set to 0.1. As a result, there are 23 complexes filtered out from the training set. We train VFNet on a single NVIDIA RTX4090 GPU, and we use the Adam as our optimizer with learning rate `1e-4`, `betas = (0.99, 0.999)`, batch size 4.

## E.3    FEATURIZATION

Whether in VFNet or in VFDiff, at the $l$-th layer, we dynamically construct the protein-ligand complex as a $k$-nearest neighbors (knn) graph based on known protein atom coordinates and current ligand atom coordinates, which is the output of the $l$-th layer. We choose $k = 32$ in VFDiff and $k = 48$ in VFNet, respectively. The protein atom features include chemical elements, amino acid types and whether the atoms are backbone atoms. The ligand atom types are one-hot vectors consisting of the chemical element types and aromatic information. The edge features are the outer products of distance embedding and bond types, where we expand the distance with radial basis functions located at 20 centers between $0\mathring{A}$ and $10\ \mathring{A}$ and the bond type is a 4-dim one-hot vector indicating the connection is between protein atoms, ligand atoms, protein-ligand atoms or ligand-protein atoms (Guan et al. (2023)).

## E.4    MODEL PARAMETERIZATION

Our VFDiff consists of 9 equivariant layers as equation 12 shows, and VFNet consists of 6 equivariant layers as show in equation 1. (The specific components of the model are shown in table 6). We

| Network | Module | Backbone | Input Dimensions | Output Dimensions | Blocks |
|---|---|---|---|---|---|
| VFNet | Protein Encoder | Linear layer | $N_P \times 27$ | $N_P \times 128$ | 1 |
| | Ligand Encoder | Linear layer | $N_M \times 13$ | $N_M \times 128$ | 1 |
| | Complex Encoder | EGNN | $(N_P + N_M) \times 128$ | $(N_P + N_M) \times 128$ | 6 |
| Diffusion Generation Network | Protein Encoder | Linear layer | $N_P \times 27$ | $N_P \times 128$ | 1 |
| | Ligand Encoder | Linear layer | $N_M \times 13$ | $N_M \times 128$ | 1 |
| | Complex Encoder | EGNN | $(N_P + N_M) \times 128$ | $(N_P + N_M) \times 128$ | 9 |

Table 6: Details of both VFNet and Vector Field-Guided Diffusion Model in our VFDiff

---

**Algorithm 1:** Training Procedure of VFDiff

---

**Input:** Protein-ligand binding dataset $\{\mathcal{P}, \mathcal{M}\}_{i=1}^N$, a learnable diffusion denoising model $\phi_{\theta 1}$, and a pretrained VFNet $\psi_{\theta 2}$

**while** $\phi_{\theta 1}$ not converged **do**
  $[\mathbf{X}_0^{\mathcal{M}}, \mathbf{X}_0^{\mathcal{P}}], [\mathbf{V}_0^{\mathcal{M}}, \mathbf{V}_0^{\mathcal{P}}] \sim \{\mathcal{P}, \mathcal{M}\}_{i=1}^N; t \sim \mathcal{U}(0, ..., T);$
  Move the complex to make CoM of protein atoms zero;
  Obtain vector field $\mathbf{F}$ from VFNet according to Equation 3;
  Perturb $\mathbf{X}_0^{\mathcal{M}}$ to obtain $\mathbf{X}_t^{\mathcal{M}}$ with shifts $S_t^{\mathcal{M}}$:
    $\epsilon_0, \epsilon_1 \sim \mathcal{N}(0, \boldsymbol{I});$
    $\tilde{\mathbf{X}}_t^{\mathcal{M}} = \sqrt{\bar{\alpha}_t}\mathbf{X}_0^{\mathcal{M}} + (1 - \bar{\alpha}_t)\epsilon_0;$
    $\mathbf{F}_t = -\nabla_{\tilde{\mathbf{X}}_t^{\mathcal{M}}} g_\theta(\tilde{\mathbf{X}}_t^{\mathcal{M}}, \mathbf{V}_0^{\mathcal{M}}, \mathbf{P})$ according to Equation 36 and 37;
    $\mathbf{S}_t^{\mathcal{M}} = \eta \cdot k_t \cdot \mathbf{F}_t;$
    $\mathbf{X}_t^{\mathcal{M}} = \sqrt{\bar{\alpha}_t}\mathbf{X}_0^{\mathcal{M}} + \mathbf{S}_t^{\mathcal{M}} + \sqrt{1 - \bar{\alpha}_t}\epsilon_1;$
  Perturb $\mathbf{V}_0^{\mathcal{M}}$ to obtain $\mathbf{V}_t^{\mathcal{M}}$:
    $g \sim \text{Gumbel}(0, 1);$
    $\log \boldsymbol{c}^{\mathcal{M}} = \log\left(\bar{\alpha}_t \mathbf{V}_0^{\mathcal{M}} + (1 - \bar{\alpha}_t)/K\right);$
    $\mathbf{V}_t^{\mathcal{M}} = \texttt{one\_hot}(\arg\max_i[g_i + \log c_i^{\mathcal{M}}]);$
  Predict $(\hat{\mathbf{X}}_{0|t}^{\mathcal{M}}, \hat{\mathbf{V}}_{0|t}^{\mathcal{M}})$ from $\phi_{\theta 1}$:
    $\hat{\mathbf{X}}_{0|t}^{\mathcal{M}}, \hat{\mathbf{V}}_{0|t}^{\mathcal{M}} = \phi_{\theta 1}([\mathbf{X}_t^{\mathcal{M}}, \mathbf{X}_0^{\mathcal{P}}], [\mathbf{V}_t^{\mathcal{M}}, \mathbf{V}_0^{\mathcal{P}}]);$
  Compute loss $L$ with $(\hat{\mathbf{X}}_{0|t}^{\mathcal{M}}, \hat{\mathbf{V}}_{0|t}^{\mathcal{M}})$ and $(\mathbf{X}_0^{\mathcal{M}}, \mathbf{V}_0^{\mathcal{M}})$;
  Update $\theta 1$ by minimizing $L$;
**end while**

---

choose to use a sigmoid $\beta$ schedule with $\beta_1 = \texttt{1e-7}$ and $\beta_T = \texttt{2e-3}$ for atom coordinates, and a cosine schedule suggested in Nichol & Dhariwal (2021) with $s = 0.01$ for atom types. We set the number of diffusion steps as 1000. We trained our model on one NVIDIA GeForce GTX 3090 GPU, and it could converge within 30 hours and 200k steps.

### E.5 TRAINING AND SAMPLING

We summarize the training and sampling procedure as Algorithms 1 and 2.

---

**Algorithm 2:** Sampling Procedure of VFDiff

---

**Input:** The protein binding site $\mathcal{P}$, the learned diffusion denoising model $\phi_{\theta 1}$, the pretrained
VFNet $\psi_{\theta 2}$(deriving vector field)$\&\ \psi_{\theta 3}$(position-tuning)
**Output:** Generated ligand molecule $\mathcal{M}$ that binds to the protein pocket $\mathcal{P}$
Sample the number of atoms $N_M$ of the ligand molecule $\mathcal{M}$ as described in Sec. 3
Move CoM of protein atoms to zero
Let $\mathbf{F}_{t-1} = \mathbf{O}$
Sample initial ligand atom coordinates $\mathbf{X}_T^{\mathcal{M}}$ and atom types $\mathbf{V}_T^{\mathcal{M}}$
**for** $t$ in $T, ..., 1$ **do**

    Predict $(\hat{\mathbf{X}}_{0|t}^{\mathcal{M}}, \hat{\mathbf{V}}_{0|t}^{\mathcal{M}})$ from $\phi_{\theta 1}$:

        $\hat{\mathbf{X}}_{0|t}^{\mathcal{M}}, \hat{\mathbf{V}}_{0|t}^{\mathcal{M}} = \phi_{\theta 1}([\mathbf{X}_t^{\mathcal{M}}, \mathbf{X}_0^{\mathcal{P}}], [\mathbf{V}_t^{\mathcal{M}}, \mathbf{V}_0^{\mathcal{P}}])$ according to Equation 12;

    Fine-tuning $\hat{\mathbf{X}}_{0|t}^{\mathcal{M}}$ with VFNet $\psi_{\theta 3}$:

        $\hat{\mathbf{X}}_{0|t}^{\mathcal{M}} = \hat{\mathbf{X}}_{0|t}^{\mathcal{M}} + 10 \cdot k_t' \cdot \texttt{norm}(f_\psi(\mathbf{M}_{0|t}^{\mathcal{M}}, \mathbf{M}_{0|t}^{\mathcal{P}}))$, where $k_t' = (1 - \bar{\alpha}_t)\alpha_t$, and $\texttt{norm}$
        $(\cdot)$ denotes unit regularization;

    Sample $\hat{\mathbf{X}}_{t-1}^{\mathcal{M}}$ from a standard diffusion process:

        $\hat{\mathbf{X}}_{t-1}^{\mathcal{M}} \sim q(\hat{\mathbf{X}}_{t-1}^{\mathcal{M}}|\hat{\mathbf{X}}_{0|t}, \mathbf{P})$, where $q(\mathbf{X}_t|\mathbf{X}_0, \mathbf{P}) = \mathcal{N}(\mathbf{X}_t^{\mathcal{M}}; \sqrt{\bar{\alpha}_t}\mathbf{X}_0, (1 - \bar{\alpha}_t)\boldsymbol{I})$;
    Calculate $\mathbf{F}_{t-1}$and $\mathbf{F}_t$:

        $\mathbf{F}_t = \mathbf{F}_{t-1}$;

        $\mathbf{F}_{t-1} = -\nabla_{\hat{\mathbf{X}}_{t-1}^{\mathcal{M}}} g_\theta(\hat{\mathbf{X}}_{t-1}^{\mathcal{M}}, \hat{\mathbf{V}}_{0|t}^{\mathcal{M}}, \mathbf{P})$ according to Equation 36 and 37;

    Sample $\mathbf{X}_{t-1}^{\mathcal{M}}$ from shifted posterior $p_\theta(\mathbf{X}_{t-1}^{\mathcal{M}}|\mathbf{X}_t^{\mathcal{M}}, \mathbf{P}, \mathbf{F}_t, \mathbf{F}_{t-1})$

        $\epsilon \sim \mathcal{N}(0, \boldsymbol{I})$;

        $\mathbf{S}_t = \eta_3 \cdot k_t \cdot \mathbf{F}_t$ according to Equation 5 (resp. $\hat{\mathbf{S}}_{t-1}$), where $k_t = (1 - \sqrt{\bar{\alpha}_t})\sqrt{\bar{\alpha}_t}$;

        $\mathbf{X}_{t-1}^{\mathcal{M}} = \frac{\sqrt{\alpha_t}(1 - \bar{\alpha}_{t-1})}{1 - \bar{\alpha}_t}\mathbf{X}_t^{\mathcal{M}} + \frac{\sqrt{\bar{\alpha}_{t-1}}(1 - \alpha_t)}{1 - \bar{\alpha}_t}\hat{\mathbf{X}}_{0|t}^{\mathcal{M}} + \frac{\alpha_t(\bar{\alpha}_{t-1} - 1)}{1 - \bar{\alpha}_t}\hat{\mathbf{S}}_t + \sqrt{\alpha_{t-1}}\hat{\mathbf{S}}_{t-1} +$
        $\frac{1 - \bar{\alpha}_{t-1}}{1 - \bar{\alpha}_t}\beta_t\epsilon$ according to Equation 11;

    Sample $\mathbf{V}_{t-1}^{\mathcal{M}}$ from posterior $p_\theta(\mathbf{V}_{t-1}^{\mathcal{M}}|\mathbf{V}_t^{\mathcal{M}}, \hat{\mathbf{V}}_{0|t}^{\mathcal{M}}, \mathbf{P})$ according to Equation 10;

**end for**

---

## F  MORE VISUALIZATION RESULTS

In this section, we present the property distributions of sampling results from different models. As shown in Figure 5, our model outperforms current diffusion-based models in terms of the QED metric. While it achieves an average performance on the SA metric, it slightly lags behind the motif-based DecompDiff, which represents a potential focus for future improvement. Regarding the docking affinity score, VFDiff demonstrates SOTA performance, with nearly 75% of the molecules achieving affinity scores within the top 50% of other models' results. Figure 6 illustrates that even in complex spatial configurations, our model is still capable of generating spatially complementary and energy-matched ligand molecules. Figure 7 additionally showcases the bond length distributions of the generated molecules compared to those in the reference.

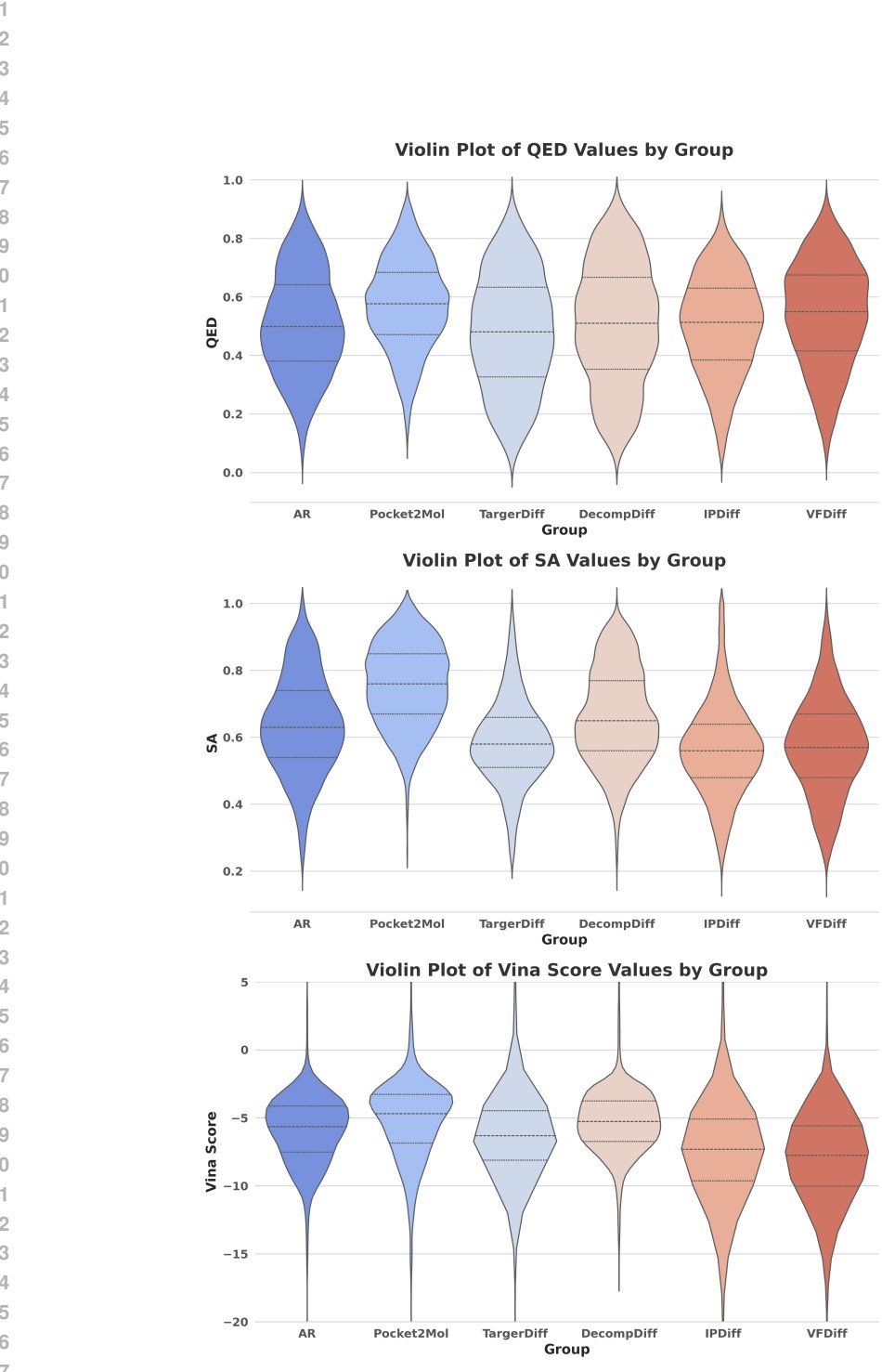

Figure 5: Generated molecules' property distribution.

## 5mma

Reference                    VFDiff

Vina: -6.8          Vina: -6.5          Vina: -6.3

## ijn2

Reference                    VFDiff

Vina: -6.3          Vina: -6.2          Vina: -6.3

## 5aeh

Reference                    VFDiff

Vina: -12.3         Vina: -14.1         Vina: -14.2

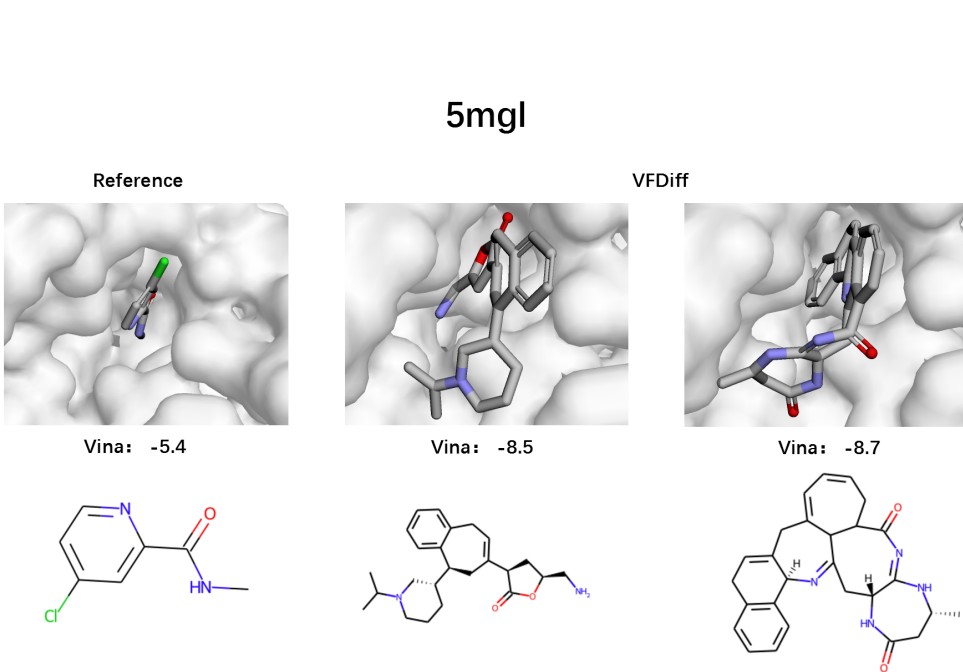

Figure 6: More examples of binding poses for generated molecules.

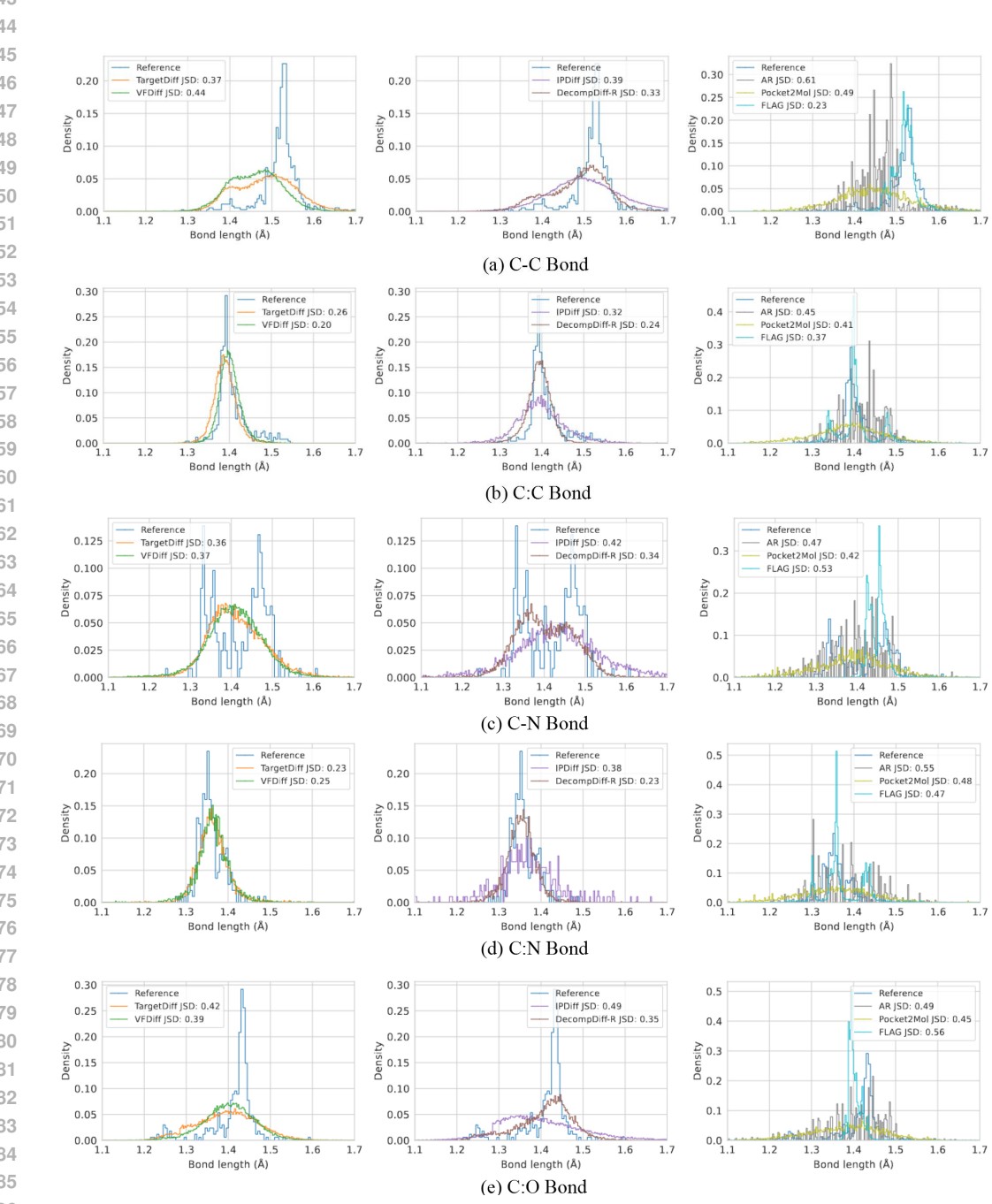

Figure 7: Bond length distribution of generated molecules compared with reference molecules

