# OpenReview forum: "VFDiff: SE(3)-Equivariant Vector Field Guided Diffusion Model for Target-Aware Molecule Generation in 3D"
_ICLR.cc/2025/Conference — Submitted to ICLR 2025_

### Official Review · Reviewer_AuiW · 2024-11-02

**Soundness:** 3
**Presentation:** 3
**Contribution:** 4
**Rating:** 8
**Confidence:** 4

**Summary:**

1.	This work introduce By incorporating energy-based guidance in both forward and reverse processes，VFDiff ensure ligand molecules are spatially complementary and energetically matched to their target pockets. VFDiff achieves the SOTA performance in SBDD scenarios.

**Strengths:**

VFDiff explicitly considers the energy complementarity between protein and ligand during the diffusion process. Therefore, it can generate three-dimensional molecules with better affinity in the process of generating protein-ligand complexes. This innovation is commendable.

**Weaknesses:**

Recently, many pocket-based three-dimensional molecular generation models have claimed to achieve state-of-the-art (SOTA) levels of affinity between protein and ligand when generating three-dimensional molecules. However, in most targets, the generated molecules often contain substructures that are difficult to synthesize, and collisions between the protein and ligand are very common. The authors did not provide enough molecular examples to demonstrate that these issues have been adequately addressed.

**Questions:**

1. When predicting forces with VFNet, it is necessary to know specific molecular information such as atom types and bond types. However, during sampling, VFDiff can only clearly provide atom types and bond types at the final step. I believe the authors need to clarify how they can provide accurate force information and position tuning information without knowing the molecular graph information in advance.
2. I look forward to the authors providing the code for reproducing the results during the review period to fairly evaluate whether the molecules generated by VFDiff have issues with synthesis and atomic collisions.
3. The authors should provide relevant results on the sampling efficiency of VFDiff.

---

### Official Review · Reviewer_6obu · 2024-11-02

**Soundness:** 3
**Presentation:** 1
**Contribution:** 1
**Rating:** 5
**Confidence:** 4

**Summary:**

The authors propose a diffusion model for structure-based drug design endowed with the affinity-based guiding strategy. To do this, authors include three components: force-guiding in diffusion process, and energy-planning and position-tuning in denoising process. Authors demonstrate that the generated molecules have a superior docking scores and perform on par with other methods in terms of other metrics.

**Strengths:**

1. I find the proposed position-tuning mechanism very interesting and original
2. The generated samples have better Vina scores than other state-of-the-art methods

**Weaknesses:**

Major:
1. The work looks very similar to IPDiff. Many parts of text, equations, figures and overall structure are almost copy-pasted from the IPDiff.
2. Technical mechanisms like prior-shifting and energy-planning are claimed to be the contributions of this paper, while they are adopted from IPDiff.
3. I believe that the style of this paper can be significantly improved:
- 3.1 The paper contains many instances of unclear language, grammar errors, and vague terminology. For instance, the meaning of the terms "spatial complementarity" (line 524) and "conditional prior" (line 144) is unclear to me.
- 3.2 The authors often use overly promotional language, which may detract from the objectivity of the work. For instance, "contributions" 2 and 3 are details of the method itself, which is already presented as contribution 1. Effectively I see two points to provide here: 1) "new" method with affinity guidance, 2) state-of-the-art performance.
4. Evaluation methodology:
- 4.1 In my opinion, some basic sanity checks like PoseBuster filters are missing. They allow to evaluate the overall adequacy of the generated molecules, checking if they are valid, connected, if the geometry of the molecules is correct, and if molecules don't have steric clashes.
- 4.2 The discussion around QED and SA metrics lacks clarity and seems inconsistent to me. If these metrics are deemed unimportant, it may be more effective to omit them from the discussion altogether. In fact, I believe that there is no reason expect these metrics to exceed the values in the training set. You are training a diffusion model which is supposed to learn the underlying distribution by design, and you are not further optimising for these metrics.
- 4.3 Since you are introducing a diffusion model which primary goal is to learn the underlying distribution, I believe that reporting JS-divergence or Wasserstein distances between training and sampled distributions of a wider range of different metrics would be helpful to understand the distribution learning capabilities of your model. For example, you can additionally compute the distances between different bond angles and atom types. Distributions of some property approximators like QED, SA, logP, etc. could also strengthen the evaluation section.
5. To me it looks like the only novel contribution of this work is position-tuning. I believe that this component can be helpful, and I like this idea in general. However, I wonder what exactly happens with $X_{0|t}$ upon position-tuning: what is the average error between $\hat{X_{0|t}^{\mathcal{M}}}$ and $X_0$ before and after position tuning in the sampling trajectories? Does it decrease as $t$ approaches to 0? Also, can you experimentally validate the hypothesis you suggest in lines 223-225? A deeper analysis of this component (potentially a separate section in Results) would be very interesting.
6. I found the explanation of position-tuning component in the end of section 4.1 very abrupt and unclear. For example, how is the position-tuning loss (4) is related to training the VFNet and losses (2)?
7. Motivation behind some design choices like introduction (as well as the choice of the values) of scaling coefficients $\eta$ are unclear to me.

**Questions:**

1. Why do you use "p" in eq 6 and "q" in eq 7?
2. Algorithm 1 (blue part): what is the difference between $\epsilon_1$ and $\epsilon_0$?
3. Algorithm 2 (blue part): why do you multiply by 10 in position-tuning?
4. Technical details: how did you assign bonds? How many molecules per input did you sample? How do you define pocket? What is the size distribution?

**Details Of Ethics Concerns:**

I find that this paper has a very similar content to IPDiff [1].

[1] Huang Z. et al. Protein-ligand interaction prior for binding-aware 3d molecule diffusion models //The Twelfth International Conference on Learning Representations. – 2024.

---

> ### Author Response · Authors · 2024-11-21
> **Response to Reviewer 6obu （Part1/2）**
>
> We sincerely thank you for your time and efforts in reviewing our paper, and your valuable feedback. We provide detailed answers in the following, and the **revised texts in manuscript are denoted in red**.
>
> **Q1: The work looks very similar to IPDiff. Technical mechanisms like force-guiding and energy-planning are claimed to be the contributions of this paper, while they are adopted from IPDiff.**
>
> **A1**: Thank you for your thoughtful feedback. We sincerely appreciate the opportunity to clarify our contributions and provide additional context regarding our work.
>
> First, controlling the generation of diffusion models using shifted-bias is a common practice in the image and text domains[1], and therefore, it is not a technique unique to IPDiff.
>
> Second, our core idea is fundamentally different from previous works. In the image domain, the primary goal is to improve generative performance by introducing a shifted-bias to modify the final distribution of the forward process, aligning it as closely as possible with the prior distribution. In IPDiff, the authors attempt to incorporate protein-ligand interactions as prior information (shifted-bias) into both the forward and backward processes of the diffusion model. However, this approach faces significant challenges, as outlined in our **A1** to **Reviewer kczy**, specifically in the **Briefings in IPDiff** and **Weakness in IPDiff** sections.
>
> In contrast, VFDiff proposes constraining docking energy changes during the forward process without altering the final distribution. Using VFNet, we compute the affinity score to derive the energy gradient and the force field acting on the molecule within the protein pocket (two representations of the same concept). During the forward process, our approach leverages docking energy guidance, where the shifted-bias corresponds to the energy gradient. We term this **energy-planning**. In the backward process, molecular formation is influenced by the protein pocket's force field, which we call **force-guiding**. Compared to IPDiff’s non-equivariant prior-shifting bias obtained through unsupervised linear transformations, our method maintains SE(3)-equivariance in molecular coordinates and invariance in molecular properties during both forward and backward processes, making it more elegant and interpretable.
>
> Third, both IPDiff and VFDiff fundamentally belong to the category of shifted-diffusion models. Structurally, we referenced these prior works, but the methods of trajectory modification via shifted-bias differ. IPDiff adopts a post-hoc modification paradigm, whereas we employ a pre-planned modification approach.
>
> Thank you again for your thorough review. We hope our response helps clarify both the similarities and unique contributions of our work compared to previous studies. Based on the points outlined above, we have revised our manuscript accordingly.
>
> [1]: Zhou Y, Liu B, Zhu Y, et al. Shifted diffusion for text-to-image generation, CVPR, 2023.
>
> **Q2: Style of the paper**
>
> **A2**: Thank you for your careful review. Based on your suggestion, we have revised the relevant statements in the paper accordingly. Additionally, we conducted a thorough review of other sections and made further improvements to enhance clarity and precision.
>
> **Q3: Details in position-tuning**
>
> **A3**: Thank you for your recognition and curiosity about the position-tuning method we proposed. We are delighted to provide a detailed explanation to address your concerns.
> *  Since our model employs the $X_0$-predict paradigm to calculate the distribution of the molecule at the previous timestep, we aim to make the predicted $X_0$ as accurate as possible. In VFDiff, the value of $(\hat{X}_{0|t} - X_0)^2$ on the validation set shows a significant negative correlation with $t$. The table below lists the mean squared error (MSE) across $t$ values ranging from 1 to 1000.
>
> |(Time) |1 | 100 | 200 | 300 | 400 | 500 | 600 | 700 | 800 | 900 | 1000 |
> |-|-|-|-|-|-|-|-|-|-|-|-|
> |(Loss)| 0.00| 0.01 | 0.01| 0.06 | 0.08 | 0.30 | 0.52|0.81 | 1.39 | 2.06 | 2.46|
>
> * To validate the hypothesis described in lines 223–225 of the paper, we conducted the following ablation experiment. We loaded the pretrained weights of the diffusion model and VFNet from VFDiff and measured the average MSE of position-tuning over the validation set under different values of scaling coefficient (called $c$ in this context). We found that the MSE is minimized when $c = 0.1$, which supports the hypothesis in lines 223–225. Beyond this value, the molecular conformation deviates significantly from the original structure. **We have already included this part of the experiment in the manuscript. Please refer to it.**
>
> |(Scaling coefficient)| $c=0$| $c=0.05$ | $c=0.1$ | $c=1$ |$c=10$|
> |-|-|-|-|-|-|
> |(Position loss)| 0.6995| 0.6973 | 0.6876| 0.7798 | 10.464|

---

> ### Author Response · Authors · 2024-11-21
> **Response to Reviewer 6obu (Part 2/2)**
>
> *  To further illustrate the impact of $c$ on the sampling results in the test set, we extended our comparative experiments. The results are summarized in the table below.
>
> |Methods| Vina Score(↓)| Vina Min (↓)| QED(↑)|  SA(↑)|
> |-|-|-|-|-|
> |c=0.1|-6.55| -7.24| 0.49|0.62|
> |c=1|-7.01|-7.96|0.52|0.58|
> |**c=10**|**-7.37** |**-8.18**|**0.54**|0.57|
> |c=20|-6.76|-7.75|0.53|0.57|
> |**Reference**|-6.36|-6.71|0.48|**0.73**|
>
> * The experiments indicate that when $c = 0.1$, the overall results are closest to the reference, further confirming the validity of our earlier hypothesis. Interestingly, the best performance was achieved at $c = 10$, despite the fact that in **table 2**, the MSE is relatively large for this value of $c$. We believe a possible explanation lies in the unordered nature of graph structures. Specifically, each node in a graph has no inherent ordering. For instance, the six carbon atoms in a benzene ring are entirely equivalent. However, when training the diffusion model to compute positional loss, we must arbitrarily assign a numbering to the carbon atoms (e.g., by traversing clockwise as 1, 2, 3, 4, 5, 6). If the same benzene ring is rotated 180 degrees counterclockwise around its center, the sequence becomes 4, 5, 6, 1, 2, 3. While these represent two entirely equivalent benzene rings in terms of molecular conformation, the positional loss calculation will yield a very large value, as each atom appears to have moved to the opposite side. From this perspective, we believe the results in tables (2) and (3) are not contradictory.
> * The average loss $| X_{i,0}-\hat{X}_{0,i}  |$  on the validation set is approximately $\eta \times \epsilon$ ($\epsilon \sim \mathbf{N}(0, I)$) when $\eta = 0.1$. To ensure generalization, we set five times this value during training VFNet.
>
> We hope this explanation helps to resolve your confusion and provides deeper insight into our position-tuning approach. Please let me know if any further clarification or adjustments are needed.
>
> **Q4: Evaluation methodology**
>
> **A4**: Thank you for your suggestion.
> *  We have added two additional metrics from PoseCheck[1], *Steric Clashes* and *Strain Energy*, to further evaluate the quality of the generated molecular conformations. The table below presents a comparison with the baselines. It can be observed that VFDiff achieves excellent performance on the *Steric Clashes* metric and shows nearly a 30% improvement in the *Strain Energy* metric compared to the other two diffusion-based models.
>
> |Methods|clashes (-)| energ (-) |
> |-|-|-|
> |TargetDiff |10.4|1410.2|
> |IPDiff|8.7| 1283.7|
> |VFDiff |9.1|1028.6|
> *  We have revised the descriptions of *QED* and *SA* in the manuscript to ensure consistency in the conveyed information.
> *  In the paper, we reported the Jensen-Shannon Divergence (JSD) of bond length distributions relative to the test set distribution. Violin plots for evaluation metrics such as *QED*, *SA*, and *Vina Score* have also been added to the **Appendix F** to better help you understand the distribution of the generated data. Additionally, we have included the JSD of bond angles (please refer to our response to **Reviewer kczy, A3**) to better illustrate the accuracy of molecular conformations.
>
> [1] PoseCheck: Generative Models for 3D Structure-based Drug Design Produce Unrealistic Poses, NeurIPS Workshop, 2023.
>
> **Q5: Why do you use "p" in eq 6 and "q" in eq 7?**
>
> **A5**: The variable $q$ describes the forward noise addition process, while $p$ describes the reverse denoising process. Therefore, $p$ in Equation (6) should be replaced with $q$. Thank you for pointing out this typo.
>
> **Q6: Algorithm 1 (blue part): what is the difference between $\epsilon_0$ and $\epsilon_1$?**
>
> **A6**: $\epsilon_0$ and $\epsilon_1$ are two independent samples drawn from a standard normal distribution. Using the reparameterization trick, we utilize $\epsilon_0$ and $\epsilon_1$ to perform sampling for the distributions in Equations (6) and (7)
>
> **Q7: Technical details: how did you assign bonds? How many molecules per input did you sample? How do you define pocket? What is the size distribution?**
>
> **A7**: We follow the paradigm of TargetDiff[2] and IPDiff[3] by generating the coordinates and types of ligand atoms, then using OpenBabel to generate the bonds. For a fair comparison, we adhere to the same settings as TargetDiff regarding the sampling of pocket positions and the number of atoms. According to the dataset statistics defined by TargetDiff, the number of atoms is positively correlated with the pocket space, and the data distribution can be described by a normal distribution.
>
> [2]: Guan et al., 3d Equivariant Diffusion for Target-aware Molecule Generation and Affinity Prediction, ICLR, 2023.
>
> [3]: Huang Z. et al. Protein-ligand interaction prior for binding-aware 3d molecule diffusion models, ICLR, 2024.
>
> If there is anything unclear or not well-explained,  further questions are welcome. We look forward to your feedback!

---

> ### Author Response · Authors · 2024-11-25
> **Thank you for your feedback**
>
> Dear Reviewer,
>
> Thank you very much for your thoughtful feedback and for recognizing our efforts to address your concerns. We greatly appreciate your adjustment in score and your acknowledgment of the improvements made, particularly regarding the new analysis of position tuning.
>
> To provide a detailed explanation of the conceptual similarities with IPDiff, we have added the following clarification:
>
> Shifted-diffusion is currently a well-established framework [1][2].  Inspired by [2], [3] was the first to apply the shifted-diffusion architecture in the SBDD (Structure-Based Drug Design) task. They attempted to incorporate protein-ligand interaction information into the noise process of ligand molecules to enhance docking performance during sampling. However, successfully applying shifted-diffusion to molecular generation is not straightforward, as IPDiff failed satisfying fundamental **geometric constraints** and ensuring **reasonable physical significance**.  Additionally, we have to pointed out that IPDiff **fully adopts** the modeling approach proposed in [2] (**noting that IPDiff neither cites [2] nor mentions this in its paper**). We believe it is necessary to highlight this to prevent readers from misjudging IPDiff's technical contributions.
> *  We would like to emphasize that our method is fundamentally based on **pocket energy perception**, which ensures the entire process maintains both equivariance and rationality—an aspect that was not achieved in previous works.
> * We proposed a novel modeling approach (a paradigm based on **pre-planned** modifications) and offered unique insights into the choice of the shifted bias (an energy-based **SE(3)-equivariant vector field**).
> * Unlike the approaches mentioned above [1][2][3], where the shifted-bias is either a constant with different scaling factors or a variable obtained through linear transformation, we proposed using the MCMC method to sample the shifted-bias at different time steps and introduced the **position-tuning** module to enhance the **sampling accuracy**.
>
> These core principles have allowed us to achieve significant advancements, with experimental results substantially surpassing those of the prior state-of-the-art (SOTA) models, including IPDiff.
>
> Additionally, we acknowledge your point regarding the stylistic similarities to IPDiff. We are committed to revising our manuscript’s overall presentation to ensure it is clearly distinct in appearance and aligns with the novel contributions of our approach.
>
> Thank you again for your valuable input, and we look forward to further refining our work in response to your suggestions.
>
> Best regards,
>
> The Authors
>
> **Reference**
>
> [1]: Zhou Y, Liu B, Zhu Y, et al. Shifted diffusion for text-to-image generation, CVPR, 2023.
>
> [2]: Zhang Z. et al. ShiftDDPMs: Exploring Conditional Diffusion Modelsby Shifting Diffusion Trajectories, AAAI, 2023
>
> [3]: Huang Z. et al. Protein-ligand interaction prior for binding-aware 3d molecule diffusion models, ICLR, 2024.

---

### Official Review · Reviewer_kczy · 2024-11-04

**Soundness:** 2
**Presentation:** 2
**Contribution:** 2
**Rating:** 5
**Confidence:** 4

**Summary:**

This paper proposes VFDiff, a novel SE(3)-equivariant diffusion model for structure-based drug design (SBDD), focusing on enhancing the binding affinity of generated molecules. VFDiff is guided by a vector field derived from VFNet, a pretrained model learning binding information by predicting atomic positions and Vina Scores from perturbed data. The authors conduct comprehensive experiments against established baselines, showing VFDiff's superior performance in generating high-affinity molecules.

**Strengths:**

- The paper is well-written and easy to follow.
- Experimental results comprehensively show the effectiveness of VFDiff in improving binding affinity.

**Weaknesses:**

- The introduction of a prior guided information in the diffusion process (i.e., energy-planning and force-guiding) is used in previous work and the authors do not point out their unique contributions relative to previous models.

**Questions:**

1. What is the novelty in Section 4.2 comparing to IPDiff?
2. What is the rationale for scaling the noise added to coordinates in VFNet training with $\eta \sim U(0, 0.5)$?
3. In Figure 10, the conformation of generated molecules appears worse than the reference ligands. Could the authors provide Jensen Shannon divergences of bond angles distributions?
4. Could the authors compare the generation efficiency of VFDiff to baseline methods?

---

> ### Author Response · Authors · 2024-11-21
> **Response to Reviewer kczy (Part1/2)**
>
> We sincerely thank you for your time and efforts in reviewing our paper, and your valuable feedback. We provide detailed answers in the following.
>
> **Q1: The unique contributions relative to previous models. And what is the novelty in Section 4.2 comparing to IPDiff?**
>
> **A1: KEY IDEA**: Inspired by the goal of drug discovery (generating small molecules with high affinity for a target), we redefine the noise addition trajectory for drug molecules. Unlike previous models, at each step of the noise addition process, the direction of the perturbation is guided by the molecular force field (Vector Field in our paper), **moving in the direction of the fastest decrease in affinity**. We demonstrate through experiments that this energy-aware trajectory outperforms previous standard processes and other path-correction methods (such as IPDiff).
>
> **Briefings in IPDiff**: The authors attempt to incorporate the relationship between protein-ligand interactions into the trajectory alterations of the forward process in their diffusion model, which is indeed a motivated approach. Specifically, they compute an interaction representation
> $F\in R^{N\times D} $
>   (where
> $N$ denotes the number of atoms and
> $D$ the dimension of the interaction representation) for each atom in the ligand. However, the ligand’s coordinate matrix
> $X\in R^{N\times 3}$ does not match the dimensions of
> $F$, making it impossible to directly manipulate the coordinates. To address this, the authors use a linear layer to reduce
> $F$ to a 3-dimensional shifted-bias vector, which they then incorporate into the control of the trajectory.
>
> It is important to note the following:
>
> **Weakness in IPDiff**: The linear transformation used is not an SE(3)-equivariant operation. As a result, although IPDiff adopts an EGNN model, the non-equivariant operation leads to ligand coordinate transformations during the forward and backward processes that do not satisfy the principles of equivariance. This is both problematic and inelegant.
>
> **Contribution in VFDiff**: In contrast, VFDiff simulates and computes an SE(3)-invariant binding energy score using VFNet, and the shifted-bias derived through differentiation maintains SE(3)-equivariance. We explain this aspect in Section 4.2.1 and provide a proof in Appendix C4. Please refer to these sections for further details.
>
> **Weakness in IPDiff**: While the protein-ligand interaction representations
> $F$ in IPDiff are meaningful (since
> $F$ is learned through a supervised docking score prediction task), the shifted-bias **s**, obtained through an unsupervised linear transformation, lacks a clear and guaranteed interpretation.
>
> **Contribution in VFDiff**: In our approach, the gradient obtained by differentiating the docking score with respect to the molecular coordinates is a force field (referred to as a “**Vector Field**” in our paper), indicating the forces exerted on the molecule within the protein pocket. Furthermore, the opposite direction of the gradient points towards the direction in which the docking score decreases. Compared to
> **s** in IPDiff, the **s** generated by our method has a tangible physical interpretation.
>
> **Q2: What is the rationale for scaling the noise added to coordinates in VFNet training with $\eta \in U(0,0.5)$?**
>
> **A2:** Thank you for your careful observation. In our paper, the diffusion model is trained using the $X_0$-predict paradigm. The average prediction error on the validation set, $|\hat{X}_{0|t,i} - X_0,i|$, where $t \in U(1,1000)$, is approximately $\eta \times \epsilon$ ($\epsilon \sim \mathbf{N}(0, I)$) when $\eta = 0.1$. To ensure generalization, we set the maximum value of $\eta$ to five times this value, i.e., 0.5, during the training of VFNet.
>
> **Q3: Could the authors provide Jensen Shannon divergences of bond angles distributions?**
>
> **A3:** Thank you for your suggestion. We have added experiments on the Jensen-Shannon divergences of bond angle distributions. Specifically, we selected the five most frequently occurring bond angles in the test set: CCC (18.1\%), C:C:C
> (16.0\%), CCO (9.5\%), C:C:N
> (5.7\%), and CCN (5.0\%). The table below presents the results, with the best value highlighted in bold.
>
> |Bond| AR|Pocket2Mol| TargetDiff|DecompDiff|IPDiff|VFDiff|
> |-|-|-|-|-|-|-|
> |CCC|0.372|0.380|0.345|**0.339**|0.392|0.399|
> |C:C:C|0.572|0.480|0.283|0.255|0.373|**0.247**|
> |CCO|0.477|0.475|0.440|**0.390**|0.490|0.473|
> |C:C:N|0.537|0.506|0.454|**0.429**|0.517|0.447|
> |CCN|0.477|0.443|0.437 |**0.418**|0.462 |0.433|

---

> ### Author Response · Authors · 2024-11-21
> **Response to Reviewer kczy (Part 2/2)**
>
> **Q4: Could the authors compare the generation efficiency of VFDiff to baseline methods?**
>
> **A4:** Thank you for your suggestion. To analyze the sampling efficiency of our model, we compared the average time consumption for generating 100 valid molecules among five baseline methods: AR, Pocket2Mol, TargetDiff, DecompDiff, and IPDiff. The results are shown in the table below, with the unit being seconds per 100 molecules. The results show that our model sacrifices a small amount of time in exchange for significant performance improvement, making this **trade-off worthwhile**
>
> | AR| Pocket2Mol| TargetDiff| DecompDiff| IPDiff| VFDiff|
> |-|-|-|-|-|-|
> |7698|2513|  3396|6174|   4284 | 4636|

---

> ### Comment · Reviewer_kczy · 2024-11-26
>
> Thanks for the authors' reply, and I do appreciate the authors’ efforts in improving IPDiff for better performance. I decide to adjust my score from 3 to 5. However, I still hold some concerns. From my point of view, although VFDiff addressed some problems of IPDiff, the overall novelty is relatively weak.
> A minor concern is that the JSD of bond distance, as reported in Table 1, is quite low for both IPDiff and VFDiff. However, when I used the generated molecules provided in the supplementary material to calculate JSD for bond, some of them are inconsistent with the result reported in the paper. It would be helpful if the author could provide the code for JSD calculation.

---

> ### Author Response · Authors · 2024-11-27
>
> Dear Reviewer kczy,
>
> We deeply appreciate your kind and thoughtful feedback, as well as your recognition of our work. Your affirmation that we have effectively addressed critical issues left by previous studies and proposed a model design for SBDD tasks from the perspectives of shape complementarity and energy matching, all while achieving outstanding performance under fundamental geometric constraints, means a great deal to us. Furthermore, we would like to emphasize the **position-tuning** module proposed in our paper. This novel component plays a crucial role in the success of the aforementioned improvements by effectively **refining the sampled data distribution**, **enhancing the accuracy of molecular conformations**, and **optimizing docking poses**. This technical component is remarkably **simple**, yet **logical** and **effective**, and we hope that this innovation will earn your recognition.（You may refer to the supplementary experiments provided in our response **A3** to **Reviewer 6obu.** and our **respond to the public comment** below)
>
> We are more than happy to address the minor concerns you have raised. TargetDiff[1] is the first work to introduce diffusion models into the structure-based drug design (SBDD) task. It defines the splits for training, validation, and test sets, as well as a series of evaluation metrics. To ensure a fair and impartial comparison of model performance, subsequent works, including but not limited to DecompDiff[2], IRDiff[3], and IPDiff[4], have claimed to follow the same setup as TargetDiff for training, testing, and evaluation. Therefore, the data presented in the tables of our paper are cited directly from the respective publications of these works (including IPDiff).
>
> The results in Tables 1 and 2 of our paper were generated using **the same evaluation code** as that used in TargetDiff. We have also uploaded the **log files** from our testing process in the supplementary materials for your review. If you have any additional questions, we are more than willing to assist and provide clarification.
>
> Once again, thank you for your time and effort.
>
> Sincerely,
>
> The Authors
>
> **Reference**
>
>
> [1]: Guan et al., 3d Equivariant Diffusion for Target-aware Molecule Generation and Affinity Prediction, ICLR, 2023.
>
> [2]: Guan et al.,DecompDiff: Diffusion Models with Decomposed Priors for Structure-Based Drug Design, ICML, 2023.
>
> [3]: Huang Z. et al. Interaction-based Retrieval-augmented Diffusion Models for Protein-specific 3D Molecule Generation, ICML, 2024.
>
> [4]: Huang Z. et al. Protein-ligand interaction prior for binding-aware 3d molecule diffusion models, ICLR, 2024.

---

> ### Public Comment · ~Zhenkun_Huang3 · 2024-11-29
> **Public Comment**
>
> Dear Reviewer,
>
> I would like to ask whether you have been able to resolve the reproducibility issue with this paper. The insights presented in this work are impressive, but I have encountered difficulties in replicating the reported performance as well. Since the authors appear not to have made their responses publicly available, I am unable to access the additional implementation details to reproduce the reported metrics. I would be deeply grateful if the authors could provide the model weights and training/inference code.
>
> Additionally, I noticed that the diffusion theory proposed in this paper is essentially the same as that of IPDiff [1], yet the authors do not seem to have explicitly acknowledged this. I believe it is necessary to properly cite the prior works in the diffusion framework in Section 4.2, as well as the theoretical proofs in the Appendix.
>
> Thank you for your time and assistance.
>
> [1] Protein-Ligand Interaction Prior for Binding-aware 3D Molecule Diffusion Models

---

> ### Author Response · Authors · 2024-11-29
>
> Hi, zhenkun
>
>  Thank you for your interest in our work. We have made our responses to the reviewers publicly available, and you can find relevant information there. Regarding the reproduction of experimental results, we have uploaded all information about the generated molecules, visualization code, and log files retained during testing in the **supplementary materials**. The data in the tables of the paper were obtained by testing with the code provided by TargetDiff. We plan to release our model code once the paper is accepted.
>
> Shifted-diffusion is currently a well-established framework and not the work of any single paper[1][2][3]. You can refer to the references we provided, as **each modeling approach differs**.  However, successfully applying shifted-diffusion to molecular generation is not straightforward, as it requires satisfying fundamental **geometric constraints** and ensuring **reasonable physical significance** (please refer to our response **A1** to reviewer **kczy**).
>
> For the marginal distribution design in the shifted-diffusion method discussed in Section 4.2, we proposed a novel modeling approach (a paradigm based on **pre-planned** modifications) and offered unique insights into the choice of the shifted bias (an energy-based **SE(3)-equivariant vector field**). Unlike the approaches mentioned above, where the shifted-bias is either a constant with different scaling factors or a variable obtained through linear transformation, we proposed using the MCMC method to sample the shifted-bias at different time steps and introduced the **position-tuning** module to enhance the **sampling accuracy**. This allows for more precise control of the offset in each denoising step, and the process adheres to SE(3)-equivariance. Regarding the proof of the formulas, we will revise the references to related work in the updated version (a minor typo in Appendix c3) and add a detailed explanation of this technique in the *Related Work* section. We hope our response helps you better understand the related works and our **unique contributions**.
>
> Thank you for your comments and support.
>
> Sincerely,
>
> The Authors
>
> **Reference**
>
> [1]: Zhou Y, Liu B, Zhu Y, et al. Shifted diffusion for text-to-image generation, CVPR, 2023.
>
> [2]: Zhang Z. et al. ShiftDDPMs: Exploring Conditional Diffusion Modelsby Shifting Diffusion Trajectories, AAAI, 2023
>
> [3]: Huang Z. et al. Protein-ligand interaction prior for binding-aware 3d molecule diffusion models, ICLR, 2024.

---

> ### Author Response · Authors · 2024-11-30
>
> Dear Reviewer kczy,
>
> On behalf of all the authors, I would like to extend our sincerest greetings to you and express our heartfelt gratitude for the time and effort you have dedicated during the review process. As the discussion phase is coming to the end, we are keen to ensure that all your concerns have been fully addressed. We sincerely look forward to your feedback and further support.
>
> Finally, we wish you a wonderful Thanksgiving holiday!
>
> Warm regards,
>
> The Authors

---

> ### Comment · Reviewer_kczy · 2024-11-30
>
> I would like to thank the authors for their responses and efforts to address my concerns. While I appreciate the potential of this paper, I still have substantial reservations about its publication in its current state.
>
> My main concern about reproducibility is partially addressed. The log file provided is consistent with the results reported in the paper. However, code necessary for reproduction is still missing.
>
> Besides, I agree with the Reviewer 6obu that the difference between VFDiff and IPDiff is small. The current manuscript and rebuttals tend to focus on favorable outcomes without adequately acknowledging the contributions and similarities to IPDiff. I believe a more detailed and honest discussion of how your work relates to prior methods would enhance the clarity and impact of your contributions.

---

> > ### Author Response · Authors · 2024-12-02
> > **To Reviewer kczy**
> >
> > Dear Reviewer kczy,
> >
> > In our previous response, we provided the code (https://anonymous.4open.science/r/TestVFDiff-E477/README.md) necessary for testing Bond JSD and added supplementary explanations in the manuscript regarding the development of shifted-diffusion and its application in SBDD tasks. We are eager to know if these updates have effectively addressed your concerns and enhanced the completeness of the paper.
> >
> > We sincerely hope for your suggestions and support.
> >
> > Thank you for your invaluable contribution to improving the quality of our work.
> >
> > Sincerely，
> >
> > The Authors

---

> > ### Author Response · Authors · 2024-12-02
> > **Follow-up on Manuscript Review**
> >
> > Dear Reviewer kczy，
> >
> > I hope this email finds you well. As the discussion phase is approaching its conclusion in just one day, we would like to check if you might have any additional concerns or suggestions regarding the manuscript. We greatly value your feedback, which plays a crucial role in ensuring the quality and impact of our work.
> >
> > If there are any outstanding points you wish to raise, we would be grateful to address them promptly. Your insights and recommendation are very much appreciated, and we look forward to your response.
> >
> > Thank you very much for your time and kind support. Please do not hesitate to let us know if there is anything further we can assist with.
> >
> > Best regards,
> >
> > The Authors

---

> > ### Author Response · Authors · 2024-12-03
> > **To Reviewer kczy**
> >
> > Dear Reviewer kczy,
> >
> > I sincerely apologize for disturbing you, but we are in great need of your support. We have done our utmost to address your concerns and implement the suggested improvements.
> >
> > We genuinely look forward to receiving your response, and on behalf of all the authors, I would like to express our heartfelt gratitude to you.
> >
> > Best regards,
> >
> > The Authors

---

> ### Author Response · Authors · 2024-11-30
> **Response to Reviewer kczy**
>
> Dear Reviewer kczy,
>
> We sincerely apologize for not being able to upload the reproduction code in a timely manner. Since the submission portal is now closed, we have uploaded the code for JSD calculation to an anonymous GitHub platform for your review: https://anonymous.4open.science/r/TestVFDiff-E477/README.md You can follow the instructions in the README file to complete the reproduction process.
>
> We greatly appreciate your suggestions regarding the discussion of our work in relation to IPDiff and will actively incorporate them. In the revised version, we will add the following discussion to help readers properly understand both our contributions and the aspects borrowed from related work.
>
> In the *Related Work* section, we will add a new subsection on **shifted-diffusion**, with the following details:
>
> [1][2] were the first to propose the shifted-diffusion framework. They designed a more effective forward process by utilizing given conditions to form a new type of conditional DDPMs, benefiting from this approach. [1] attempted to control the noise trajectories of different digits on the MNIST dataset to improve the quality of generation. [2], a multimodal model, introduced textual conditional information into the image noise process, thereby altering the prior distribution during sampling and improving sampling efficiency.
>
> Inspired by [1], [3] was the **first to apply** the shifted-diffusion architecture in the SBDD (Structure-Based Drug Design) task. They attempted to incorporate protein-ligand interaction information into the noise process of ligand molecules to enhance docking performance during sampling.
>
> While [3] achieved promising results, there remain two major issues:
> 1. The introduced information must be reduced to 3D via linear transformation, which violates the fundamental geometric equivariance constraints in molecular generation.
> 2. The interpretability of the unsupervised dimensionality reduction process is problematic. The meaning of the reduced shifted-bias cannot be clearly explained, limiting its interpretability.
>
> Building on the insights from [1][2][3], we propose VFDiff, a 3D equivariant molecular generation framework guided by energy transformations, to address these challenges. Starting from the design concept of **energy matching**, we have re-examined the shifted-diffusion framework. We believe that the noise trajectory of molecules should shift in the direction that maximizes the increase in binding energy, while also satisfying the requirement of equivariance (this is our **conceptual contribution**). Next, we present our **technical contributions**. In the marginal distribution design, unlike the guiding approaches of [1][2][3], we introduce a pre-plan guiding paradigm, where guidance is incorporated prior to data scaling. For guidance computation, we propose using an MCMC sampling method optimized by **position-tuning** to compute the vector field between molecular conformations at different time steps and the protein pocket. This ensures more precise guidance during the sampling process and more guaranteed interpretation.
>
> We have discussed the development of shifted-diffusion and the unique contributions of each paper in the aforementioned *Related Work* section with a **honest and impartial attitude**. Additionally, we have pointed out that IPDiff **fully adopts** the modeling approach proposed in [1] (**noting that IPDiff neither cites [1] nor mentions this in its paper**). We believe it is necessary to highlight this to prevent readers from misjudging IPDiff's technical contributions.
>
> Thank you again for your valuable suggestions, and we wish you a wonderful Thanksgiving!
>
> Sincerely，
>
> The Authors
>
> **Reference**
>
> [1]: Zhang Z. et al. ShiftDDPMs: Exploring Conditional Diffusion Modelsby Shifting Diffusion Trajectories, AAAI, 2023
>
> [2]: Zhou Y, Liu B, Zhu Y, et al. Shifted diffusion for text-to-image generation, CVPR, 2023.
>
> [3]: Huang Z. et al. Protein-ligand interaction prior for binding-aware 3d molecule diffusion models, ICLR, 2024.

---

> > ### Public Comment · ~Zhenkun_Huang3 · 2024-12-01
> > **Inquiries about Reproducibility Issues**
> >
> > Dear Authors,
> >
> > I sincerely appreciate the time and effort the authors have dedicated to providing the code for calculating the metrics. However, it appears that the provided code only supports metric calculation and is derived from the TargetDiff repository, rather than being suitable for reproducing the generated results. Since the reproducibility of generated results is a critical aspect in evaluating generative models, I would be grateful if the authors could further provide the model weights and the training/inference code to enhance the reproducibility of this work. When I attempted to replicate the work following the details described in the paper, I was unsuccessful, possibly due to the absence of certain technical details.

---

> ### Author Response · Authors · 2024-12-01
>
> Dear Reviewer kczy,
>
> The code we prepared has been fully debugged and is now available for your review. We look forward to your further feedback and suggestions and will respond to your inquiries promptly. Thank you for your time and effort.
>
> Sincerely,
>
> The Authors

---

> ### Author Response · Authors · 2024-12-01
>
> Hi, zhenkun,
>
> In our manuscript, we have explicitly committed to releasing the code and model weights promptly after the paper is accepted. The provided code is intended to address the reviewers' concerns. If you are interested in our work, we would be happy to discuss it with you after the review process concludes. Thank you for your comments.

---

> ### Public Comment · ~Zhenkun_Huang3 · 2024-12-01
> **Public Comment about Reproducibility Issues**
>
> Dear Authors,
>
> Thank you for all your responses. However, I still have concerns regarding the reproducibility of generated results and reported performance in VFDiff, as I encountered difficulties when attempting to replicate the results, and the training or inference code has not been provided. I am looking forward to the release of the code and model weights in the future, as this would grately contribute to the advancement of molecule generation.

---

> > ### Author Response · Authors · 2024-12-01
> >
> > Thank you for your understanding. Please be patient as we are working hard to organize the code and provide detailed annotations.
> >
> > We promise to release the complete code along with comprehensive annotations after the paper is accepted. We look forward to receiving more of your valuable feedback at that time.

---

> ### Comment · Reviewer_kczy · 2024-12-03
>
> I appreciate the effort of the authors in addressing my concerns. After checking the evaluation code the authors provided, I came to know that the problem is not introduced by the authors. Instead, it is the reliance on the reference distribution provided by TargetDiff, which differs from the true distribution of the CrossDocked2020 test set, leads to a consistent underestimation of the JSD values for all methods. For example, the bond JSD of 'C=C' between Pocket2Mol and this reference distribution is 0.292. To ensure the results accurately reflect the true performance of the models, I strongly recommend recalculating the reference distribution based on the CrossDocked2020 test set and reevaluating the bond and angle JSD metrics.
>
> Unfortunately, the preliminary results shared by the authors, along with this corrected perspective, deepen my concerns about the rationality of the generated conformations. The visualization in Figure 6 also suggest that the generated poses may lack structural coherence or alignment with expected physical and chemical constraints. To address this comprehensively, a more thorough analysis of the validity of the generated molecular conformations is required. Employing tools such as PoseCheck [1] or PoseBuster [2] could provide valuable insights. I look forward to seeing these additional evaluations and updates in future revisions.
>
> [1] Harris, C., Didi, K., Jamasb, A., Joshi, C., Mathis, S., Lio, P., & Blundell, T. (2023). Posecheck: Generative models for 3d structure-based drug design produce unrealistic poses. In NeurIPS 2023 Generative AI and Biology (GenBio) Workshop.
>
> [2] Buttenschoen, M., Morris, G. M., & Deane, C. M. (2024). PoseBusters: AI-based docking methods fail to generate physically valid poses or generalise to novel sequences. Chemical Science, 15(9), 3130-3139.

---

> > ### Author Response · Authors · 2024-12-03
> >
> > Dear Reviewer kczy:
> >
> > Thank you for pointing out the critical issues. We will reassess bond JSD and other metrics and include the calculation code. We will reply to you at the earliest possible moment once the results are available. We sincerely appreciate your meticulous review.
> >
> > Sinserely,
> >
> > The Authors

---

> > ### Author Response · Authors · 2024-12-03
> > **The updated results**
> >
> > Dear Reviewer kczy,
> >
> > Thank you very much for pointing out this critical issue. We have recalculated the Bond JSD and Angle JSD based on the data distribution in the CrossDock2020 test set, focusing on the eight most frequent bond types and the five most frequent bond angles in the test set. The notebook with the saved results has been uploaded: [https://anonymous.4open.science/r/Test-E48F/README.md](https://anonymous.4open.science/r/Test-E48F/README.md). The results are presented in the table below.
> >
> > **Bond JSD:**
> >
> > |Bond/frequency| AR|Pocket2Mol| TargetDiff|DecompDiff|IPDiff|VFDiff|
> > |-|-|-|-|-|-|-|
> > |CC (29.6%)|0.61|0.49|0.37|**0.33**|0.39|0.44|
> > |C:C(20.7%)|0.45|0.41|0.26|0.24|0.32|**0.20**|
> > |CO(13.9%)|0.49|0.45|0.42|**0.35**|0.49|0.39|
> > |CN(10.1%)|0.47|0.42|0.36|**0.34**|0.42|0.37|
> > |C:N(8.8%)|0.55|0.48|0.23 |**0.23**|0.38 |0.25|
> > |OP(4.5%)|0.53|0.81|0.44|**0.45**|0.48 |0.47|
> > |C=O(4.2%)|0.56|0.51|0.46 |**0.39**|0.43 |0.45|
> > |C=C(1.7%)|0.56|0.54|**0.50**|0.54|0.57 |0.52|
> >
> > **Angle JSD**
> >
> > |Bond/frequency| AR|Pocket2Mol| TargetDiff|DecompDiff|IPDiff|VFDiff|
> > |-|-|-|-|-|-|-|
> > |CCC(18.1%)|0.372|0.380|0.345|**0.339**|0.392|0.399|
> > |C:C:C(16.0%)|0.572|0.480|0.283|0.255|0.373|**0.247**|
> > |CCO(9.5%)|0.477|0.475|0.440|**0.390**|0.490|0.473|
> > |C:C:N(5.7%)|0.537|0.506|0.454|**0.429**|0.517|0.447|
> > |CCN(5.0%)|0.477|0.443|0.437 |**0.418**|0.462 |0.433|
> >
> > In our response A4 to Reviewer 6obu, we tested the results of **PoseCheck** (this test took a considerable amount of time, so we were unable to test more baseline methods). The results are shown in the table below:
> >
> > |Methods|clashes (-)| energ (-) |
> > |-|-|-|
> > |TargetDiff |10.4|1410.2|
> > |IPDiff|8.7| 1283.7|
> > |VFDiff |9.1|1028.6|
> >
> >
> > Based on the above test results, we believe that VFDiff does not have issues with inaccurate molecular conformations. However, as the testing method has changed, we will adjust the corresponding descriptions of the results in the paper for the sake of scientific rigor and reselect the examples for presentation.
> >
> > Finally, on behalf of all the authors, please allow me to express our heartfelt gratitude to you. Your meticulous and rigorous scientific attitude and the suggestions you provided are invaluable not only to us but also to other members of the molecular generation community. Thank you for your dedication during this period.
> >
> > Sincerely,
> >
> > The Authors

---

> ### Author Response · Authors · 2024-12-04
> **Supplementary Experiments on Torsion Angle JSD**
>
> Dear Reviewer kczy,
>
> To thoroughly address your concerns regarding the reasonableness of molecular conformations, we have added the distributions of the top-8 most frequent torsion angles between the reference and the generated molecules, as shown in the table below.
>
> **Torsion Angle JSD:**
>
> |Bond/frequency| AR|Pocket2Mol| TargetDiff|DecompDiff|IPDiff|VFDiff|
> |-|-|-|-|-|-|-|
> |CCCC(13.7%)|0.38|0.32|**0.31**|0.35|0.32|0.34|
> |C:C:C:C(12.5%)|0.71|0.53|0.37|**0.24**|0.48|0.37|
> |CCOC(5.3%)|0.37|0.37|0.34|0.34|0.35|**0.34**|
> |CCCO(5.0%)|0.43|0.40|0.40|0.41|0.41|**0.40**|
> |CCNC(4.7%)|0.41|0.43|0.41 |**0.39**|0.43 |0.42|
> |C:C:N:C(4.7%)|0.68|0.54|0.42 |**0.23**|0.47 |0.40|
> |C:C:C:N(3.1%)|0.66|0.53|0.45 |**0.31**|0.52 |0.43|
> |C:N:C:N(2.9%)|0.78|0.55|0.49 |**0.27**|0.51 |0.48|
>
> Combined with Bond JSD and Angle JSD, it is evident that diffusion models significantly outperform autoregressive models in terms of conformation accuracy. Among diffusion models, DecompDiff, which is based on fragment generation, clearly outperforms other diffusion models.
>
> Overall, our proposed VFDiff achieves the second-best performance in terms of conformation reasonableness among diffusion models, behind DecompDiff, while being on par with TargetDiff (and significantly better than IPDiff). Most importantly, the primary goal of our model design is to improve molecular **docking affinity**, where we demonstrate a substantial performance advantage. Compared to IPDiff, DecompDiff, and TargetDiff, our model shows improvements of **15**%, **30**%, and **35**%, respectively, in the Vina Score metric.
>
> Therefore, we believe that the proposed energy-based shifted-diffusion and position-tuning mechanisms are of significant importance.
>
> Once again, thank you for your insightful comments and continued support.
>
> Sincerely,
>
> The Authors

---

### Meta-Review · Area_Chair_rAFx · 2024-12-22

**Metareview:**

This paper proposes VFDiff, a diffusion model for structure-based drug design that incorporates energy-based guidance in both forward and reverse processes. The core technical contributions include a vector field guidance mechanism derived from protein-ligand binding energy, position-tuning during sampling to improve conformational accuracy, and energy-planning and force-guiding components to ensure molecules are both spatially and energetically matched to target pockets. Experiments on CrossDocked2020 demonstrate improved binding affinity scores compared to prior methods.

The work has several strengths, including a novel position-tuning mechanism that provides meaningful improvements in sampling accuracy, strong empirical results on docking scores and binding affinity metrics, clear experimental validation including detailed ablation studies, and detailed responses to reviewer concerns with additional experiments and analyses.

However, significant weaknesses emerged during the review process. First, the technical novelty compared to prior work (particularly IPDiff) appears limited. As noted by Reviewer 6obu, core mechanisms like force-guiding and energy-planning appear largely adopted from IPDiff. While the authors' rebuttal acknowledges using the shifted-diffusion framework from prior work, the differentiation from IPDiff remains unclear despite their explanations. Second, there are serious reproducibility concerns - Reviewer kczy identified issues with bond/angle JSD calculations using reference distributions, code for training/inference was not provided during review, and public comments highlight difficulties reproducing key results. Third, the evaluation was incomplete, lacking important metrics like PoseCheck for validating molecular conformations initially. Bond angle distributions had to be recalculated after reviewer feedback, and questions remain about synthetic accessibility of generated molecules.

Based on these points, paper's primary technical contribution appears incremental over IPDiff - while the position-tuning mechanism is novel, this alone does not constitute sufficient advancement for publication. Multiple reproducibility concerns emerged during review that were not fully resolved, including discrepancies in reported metrics, lack of training/inference code, and unaddressed questions about result replication. Finally, the initial evaluation omitted important baseline comparisons and metrics, suggesting incomplete validation of the method's capabilities.

While the authors provided detailed responses and additional experiments, the core issues around novelty and reproducibility remain. I recommend the authors: more clearly differentiate their technical contributions from IPDiff, provide complete code and model weights for reproduction, expand evaluation to comprehensively validate molecular quality, and consider combining with additional novel technical components. The paper shows promise but requires substantial revision to meet ICLR's standards for technical novelty and reproducibility.

**Additional Comments On Reviewer Discussion:**

Reviewers raised major concerns about technical novelty, reproducibility, and evaluation completeness.

One reviewer expressed serious concerns about similarity to IPDiff, particularly in the shifted-diffusion framework and energy guidance mechanisms. The authors responded by clarifying their novel contributions: energy-based SE(3)-equivariant vector fields, pre-planned modification paradigm, and MCMC sampling with position-tuning. While this explanation partially addressed originality concerns, the reviewer maintained that differences from IPDiff remained minimal.

The discussion around reproducibility was particularly notable. Multiple reviewers and public commenters highlighted difficulties in replicating results. The authors provided code for metric calculations and additional experimental results, including recalculated bond/angle JSD distributions and PoseCheck evaluations. However, they did not release complete training/inference code during the review period, citing plans to do so post-acceptance. This decision significantly impacted the ability to fully validate their claims.

Regarding evaluation methodology, reviewers requested additional metrics and analyses. The authors responded with comprehensive new experiments, including PoseCheck metrics for structural validation, expanded JSD calculations for bond angles and torsions, and detailed ablation studies of the position-tuning mechanism. These additions strengthened the empirical validation but also revealed some inconsistencies with initially reported metrics.

The position-tuning mechanism was agreed as the paper's most novel contribution through the discussion. The authors provided detailed analyses showing its effectiveness, including ablation studies with different scaling coefficients and their impact on molecular conformations, which was appreciated by the reviewers.

In weighing these points for the final decision, the incomplete code release and remaining questions about reproducibility were particularly concerning. While the authors made substantial efforts to address reviewer comments with additional experiments, the core issues of technical novelty relative to IPDiff and result verification remained unresolved. The strong experimental results and novel position-tuning mechanism were positive factors, but ultimately insufficient to overcome these fundamental concerns for acceptance at ICLR.

---

### Decision · Program_Chairs · 2025-01-22

Reject